https://doi.org/10.1038/s41467-021-21881-2　　**OPEN**

# The amyloid structure of mouse RIPK3 (receptor interacting protein kinase 3) in cell necroptosis

Xia-lian Wu[1,2,3,5], Hong Hu[1,2,3,5], Xing-qi Dong[1,2,3], Jing Zhang[1,2,3], Jian Wang[1], Charles D. Schwieters [4], Jing Liu[1,2,3], Guo-xiang Wu[1,2,3], Bing Li[1], Jing-yu Lin[1,2,3], Hua-yi Wang [1✉] & Jun-xia Lu [1✉]

RIPK3 amyloid complex plays crucial roles during TNF-induced necroptosis and in response to immune defense in both human and mouse. Here, we have structurally characterized mouse RIPK3 homogeneous self-assembly using solid-state NMR, revealing a well-ordered N-shaped amyloid core structure featured with 3 parallel in-register β-sheets. This structure differs from previously published human RIPK1/RIPK3 hetero-amyloid complex structure, which adopted a serpentine fold. Functional studies indicate both RIPK1-RIPK3 binding and RIPK3 amyloid formation are essential but not sufficient for TNF-induced necroptosis. The structural integrity of RIPK3 fibril with three β-strands is necessary for signaling. Molecular dynamics simulations with a mouse RIPK1/RIPK3 model indicate that the hetero-amyloid is less stable when adopting the RIPK3 fibril conformation, suggesting a structural transformation of RIPK3 from RIPK1-RIPK3 binding to RIPK3 amyloid formation. This structural transformation would provide the missing link connecting RIPK1-RIPK3 binding to RIPK3 homo-oligomer formation in the signal transduction.

[1] School of Life Science and Technology, ShanghaiTech University, Shanghai, PR China. [2] CAS Center for Excellence in Molecular Cell Science, Shanghai Institute of Biochemistry and Cell Biology, Chinese Academy of Sciences, Shanghai, PR China. [3] University of Chinese Academy of Sciences, Beijing, PR China. [4] Laboratory of Imaging Sciences, Office of Intramural Research, Center for Information Technology, National Institutes of Health, Bethesda, MD, USA. [5] These authors contributed equally: Xia-lian Wu, Hong Hu. ✉email: wanghuayi@shanghaitech.edu.cn; lujx@shanghaitech.edu.cn

Amyloids, commonly associated with the neurodegenerative disease, have been found playing extraordinary roles in various biological systems, functioning beyond neurodegeneration[1]. In necroptosis, a group of signaling complexes, termed necrosomes[2], have been characterized having structural properties as amyloids[3]. RIPK3 is the major component in the necrosomes and an indispensable player in necroptosis[4]. RIPK3 consists of a N-terminal kinase domain and a C-terminal receptor-interacting protein homotypic interaction motif (RHIM)[5]. RHIM contains the most conserved tetrad sequence I (V) QI (V) G, plus the hydrophobic sequences flanked on both sides of the tetrad (Fig. 1a). In RIPK3-mediated necroptosis, RIPK3 assembles into amyloids to provide a scaffold for its N-terminal kinase domain to self-phosphorylate or phosphorylate the downstream effector molecule mixed lineage kinase domain-like protein (MLKL)[3,6]. It is RHIM and the nearby sequence that mediates the assembly of RIPK3 amyloid oligomers[3], while the kinase domain self-phosphorylates and phosphorylates MLKL only when RIPK3 oligomerizes[7]. Upon phosphorylation, MLKL is activated and oligomerizes to disrupt the plasma membrane and cause cell necrosis[8–10].

Many proteins[11–13] in necroptosis pathway contain RHIMs (Fig. 1a), and the intermolecular RHIM–RHIM interactions are mainly responsible for the intermolecular interactions and the amyloid formation in the different necrosomes. Besides that, viral protein M45 was reported to repress RIPK3 activation through forming inhibitory amyloids with RIPK3[14,15]. We are still lacking detailed structural information on these different high-order RHIM complexes. A hetero-amyloid structure of human RIPK1/RIPK3, related to the necrosome formed in tumor necrotic factor (TNF)-induced necroptosis pathway, has been solved recently by solid-state NMR (SSNMR) (pdb:5V7Z Fig. 1b)[16]. It has been proposed that the hetero-amyloid formed upstream of RIPK3 amyloid could serve as a template to recruit free RIPK3 into the same structure, thereby transducing and amplifying necrotic signals from RIPK1 to RIPK3[3]. However, the RIPK1/RIPK3 hetero complex cannot induce RIPK3 kinase activation directly[17], the RIPK3 self-assembly is necessary to induce cell necroptosis

either downstream of RIPK1–RIPK3 binding or independently without RIPK1. It would be interesting to see the structural details of the pure RIPK3 amyloid and whether it can adopt structures similar to those of RIPK1/RIPK3 complexes.

We characterized the amyloid structure of mouse RIPK3 using SSNMR. We identified RIPK3 RHIM region consisting of three β-strands, with the conserved VQIG sequence contributing to the center β-strand. The three β-strands are arranged as an "N" shape with the last β-strand as a part of a long tail. There is only one protein molecule in each cross-β unit of mouse RIPK3 fibril, interacting with neighboring molecules in a parallel in-register fashion. This conformation is different from the hetero-amyloid structure of human RIPK1/RIPK3, although both structures show the RHIM conserved tetrad sequence as the center segment for the amyloid β-sheet structure. The hetero-amyloid contains two molecules in a single cross-β unit, with the amyloid core structure stabilized by the intermolecular interactions between the hydrophobic residue side chains mainly from the tetrad sequence of 2 molecules. However, in the mouse RIPK3 amyloid, the first two β-strands form the "β-arches" structure. The amyloid core structure is stabilized by the intramolecular interactions between the hydrophobic residue side chains from the different β-strands within the same protein molecule. Segmental (4-alanine) replacement of mouse RIPK3 on the first two β-strands individually would totally block the interaction between RIPK1 and RIPK3, and inhibit mouse NIH-3T3 cell necroptosis. On the other hand, single-site mutations at each of the three β-strands (F442D, Q449D, or L456D) of mouse RIPK3 inhibit mouse NIH-3T3 cell necroptosis while RIPK1 and RIPK3 interactions could still be maintained. And these RIPK3 mutants still form the fibril in solution. Combining these results, we could propose that both RIPK1–RIPK3 binding and the RIPK3 amyloid formation are essential, but not sufficient for TFNα-induced necroptosis. The structural integrity of RIPK3 RHIM which consists of three β-strands is necessary for RIPK3 function. To further explore the interaction mechanism between mouse RIPK1 and RIPK3, a molecular dynamics (MD) simulation was carried out on a hetero-amyloid model using mouse RIPK3 fibril structure as the template

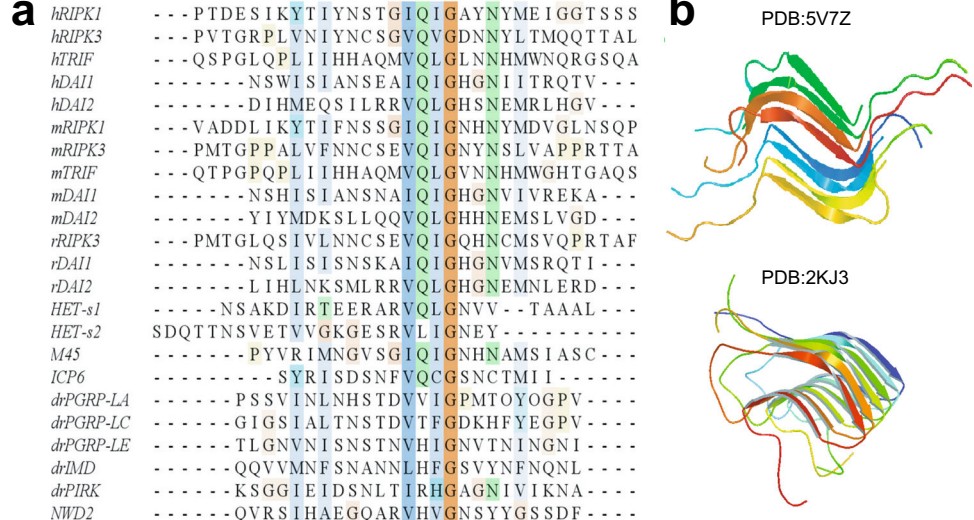

**Fig. 1 Alignment of RHIM sequence from various proteins and two published structures containing of RHIM. a** Alignment of RHIM sequence of human (h), mouse (m), rat (r), drosophila (dr), and herpesviruses (M45, ICP6) proteins and prion-forming domain of *P. anserine* proteins (Het-s1, Het-s2). The most conserved tetrad sequence is highlighted at the center. The alignment was performed by clustalw2 online software. Receptor interacting protein kinase 1 (RIPK1), toll-interleukin-1 receptor domain-containing adapter protein inducing interferon beta (TRIF), and DNA-dependent activator of interferon-regulatory factors (DAI) are three proteins found in necroptosis pathway. **b** The SSNMR structures of human RIPK1/RIPK3 hetero-amyloid (PDB:5V7Z) and Het-s fibrils (PDB:2KJ3).

but replacing half of the molecules to mouse RIPK1 sequence. While mouse RIPK3 homo-amyloid develops a twist during the simulation, the hypothetical hetero-amyloid exhibits an opening of the "β-arches" formed by the first two β-strands, showing a structure with great similarities to the human RIPK1/RIPK3 hetero-amyloid. The results indicate that lower stability for a hetero-amyloid to adopt the conformation as mouse RIPK3 homo-amyloid. These findings suggest the mouse RIPK1/RIPK3 hetero-amyloid would undergo a structural transformation from the hetero-oligomer formed upon RIPK1–RIPK3 interaction to the RIPK3 self-assembly if the structure of the hypothetical hetero-amyloid is similar to the real mouse RIPK1/RIPK3 hetero-amyloid. Amyloids are used to be considered as irreversible structural assemblies with high stability. Our results provide a picture of an amyloid structural transformation in the necroptosis signal transduction pathway from RIPK1–RIPK3 binding to RIPK3 self-assembly, explaining why RIPK1/RIPK3 hetero complex could not induce RIPK3 kinase activation directly.

## Results

**Fibril formation by mouse RIPK3.** The full-length mouse RIPK3 contains a kinase domain connecting a RHIM motif through a random coil linker as shown in Fig. 2a. The most conservative sequence [448]VQIG[451] for RHIM contributes to RIPK3 fibril architecture directly[3]. Online prediction on RIPK3 amylogenic regions using Waltz (http://waltz.switchlab.org)[18] showed a small segment ([449]QIGNYNSLV[457]) in the C-terminal domain would tend to form aggregation. However, PSIPRED (bioinf.cs.ucl.ac.uk/psipred/)[19,20] predicted the secondary structure of mouse RIPK3 C-terminal domain-containing 3 sequential segments [440]LVFN[443], [447]EVQI[450], and [455]SLV[457] that could form β-strand and may contribute to fibril formation (Supplementary Fig. 1a). Finally, a sequence from mouse RIPK3 residue 409 to the end, a 78-residue sequence covering the predicted β-strand region, was constructed (Fig. 2b) with a 6 × His tag attached at the N-terminus. Protein was expressed in *E. coli* system and purified according to the method described below.

We prepared the fibrils by dialysis of the denatured protein in $H_2O$ at room temperature. Negatively stained fibrils are relatively straight, unbranched and single-stranded showed by transmission electron microscopy (TEM) (Fig. 2c). The X-ray diffraction image of fibril in Fig. 2d indicates a typical feature of cross-β amyloid structure with equatorial and meridional diffraction at about 9.7 Å and 4.7 Å, respectively. Atomic force microscope (AFM) images (Fig. 2e, f) reveal uniform fibril heights of 1.8 ± 0.2 nm for mouse RIPK3. Measuring mass-per-length (MPL) value of fibrils is usually an effective means of characterizing the quantity of monomer in the cross-β unit of a fibril. We used dark-field Beam Tilted (BT)-TEM to obtain MPL value for the fibril[21]. A typical image shown in Fig. 2g contains a mixture of mouse RIPK3 fibrils and tobacco mosaic virus (TMV) where TMV serves as an internal standard with an MPL value of 131 kDa/nm. The MPL values of mouse RIPK3 fibrils were summarized into a histogram, fit with a Gaussian distribution in Fig. 2h. In our results, we observed a center value about 17.2 ± 0.2 kDa/nm. When one mouse RIPK3 monomer with molecular weight 9.57 kDa lies in a single cross-β unit with a spacing between cross-β units about 4.7–4.8 Å, the expected MPL value is about 19.9–20.3 kDa/nm as indicated in vertical red line in Fig. 2h. The result indicates mouse RIPK3 fibril is onefold symmetry structure across the fibril axis. The width of the Gaussian distribution (full-width-at-the-half-maximum) is 13.2 kDa/nm, caused by the background intensity variations in the images. The background intensity variations are displayed in Supplementary Fig. 2.

**Identification of the amyloid fibril core region.** In order to identify the amyloid fibril core region, SSNMR experiments on the uniformly labeled mouse RIPK3 fibrils were conducted. Cross-polarization based SSNMR experiments would reveal the immobilized structure contributed mostly from the fibrils[3,16,22]. Figure 3 shows 2D $^{13}C$–$^{13}C$ DARR, $^{13}C$–$^{15}N$ NcaCX, and $^{13}C$–$^{15}N$ NcoCX spectra with 50 ms mixing time. The linewidth for $^{15}N$ is about 1.5 ppm; the linewidth for $^{13}C$ is about 0.7–0.9 ppm. There are less peaks than expected for a 86-residue peptide, indicating only part of the sequence is involved in the fibril formation. J coupling based $^1H$–$^{13}C$ INEPT and TOBSY experiments were also carried out, showing many peaks from the peptide mobile component (Supplementary Fig. 3).

The SSNMR 2D spectra in Fig. 3 have good resolutions, showing a single morphology for the fibrils and allowing us to do the sequential assignment. 3D $^{13}C$–$^{15}N$ NCACX and 3D $^{13}C$–$^{15}N$ NCOCX spectra were also collected to facilitate the assignment. All the NMR experiments were summarized in Supplementary Table 1. We confirmed that the rigid segment that was able to be sequentially assigned was from residue 441 to 461, while most residues at the flanking region had no SSNMR signals. The chemical shifts were summarized in Supplementary Table 2. Figure 4b showed the secondary chemical shift ($\Delta\delta C\alpha - \Delta\delta C\beta$) plot where the measured chemical shift values of residue Cα and Cβ were compared to the theoretical values for a random coil structure. A negative value of $\Delta\delta C\alpha - \Delta\delta C\beta$ suggests β-sheet secondary structure. Three segments from residue V441 to N444, residue V448 to N452 and residue N454 to P460 showed negative values, indicating three β-strands. Using the TALOS-N server[23], the protein torsion angles ψ and φ were also predicted, confirming β-sheet secondary structure of the fibril (Fig. 4c). Therefore, mouse RIPK3 fibril is composed of three β-strands with the most conserved RHIM tetrad sequence as the center β-strand, generally consistent with the prediction from the online software PSIPRED (Fig. 4a and Supplementary Fig. 1a).

**Fibril structural model for mouse RIPK3.** The final structures of mouse RIPK3 were calculated using the Xplor-NIH program, with experimental distance and dihedral restraints. In particular, the inter-residue interactions were obtained using 2D $^{13}C$–$^{13}C$ correlation spectra with various mixing time from 25 to 500 ms (Fig. 5a, b, d) and $^{15}N$–$^{13}C$ z-filtered TEDOR spectrum (Fig. 5c). Sparsely labeled proteins using [1, 3-$^{13}C$]- and [2-$^{13}C$]-labeled glycerol as carbon source were used to simplify the assignment. The inter-residue correlations between the side chains of L456 and Q449 (Figs. 5a, 4c), F442 and I450 (Fig. 5b), G451 and N454 (Fig. 5c), as well as V441 and G451 (Fig. 5d) are shown in SSNMR spectra in Fig. 5 as some examples. A total of ten unambiguous nonsequential inter-residue correlations V441Cγ−G451Cα, S446Cβ−V448Cβ, Q449Cδ−L456Cβ, Q449Cδ-L456Cγ I450Cγ2−N452Cα, G451Cα−N454Nδ2, N452Cα−N454Cα, N452Cβ−N454Cα, Y453Cβ−S455Cβ, and L456Cγ−Q449Nε2 were obtained. From MPL measurements, we concluded that there was a single protein molecule in a cross-β unit of the fibril. Therefore, the inter-residue contacts obtained from SSNMR correlation spectra were all assumed to be intramolecular interactions. A RIPK3 fibril sample with equimolar-mixed $^{13}C$-labeled and $^{15}N$-labeled protein was also prepared. The 2D $^{13}C$–$^{13}C$ correlation spectra with various mixing time from 25 to 500 ms were obtained and compared to the unformly $^{13}C$, $^{15}N$-labled sample, confirming the intramolecular correlations (Supplementary Fig. 4).

The information on the intermolecular arrangement of β-strands was obtained using 2D $^{13}C$–$^{13}C$ spectra on the sample with [2-$^{13}C$]-glycerol labeling (Supplementary Fig. 5). With 50 ms

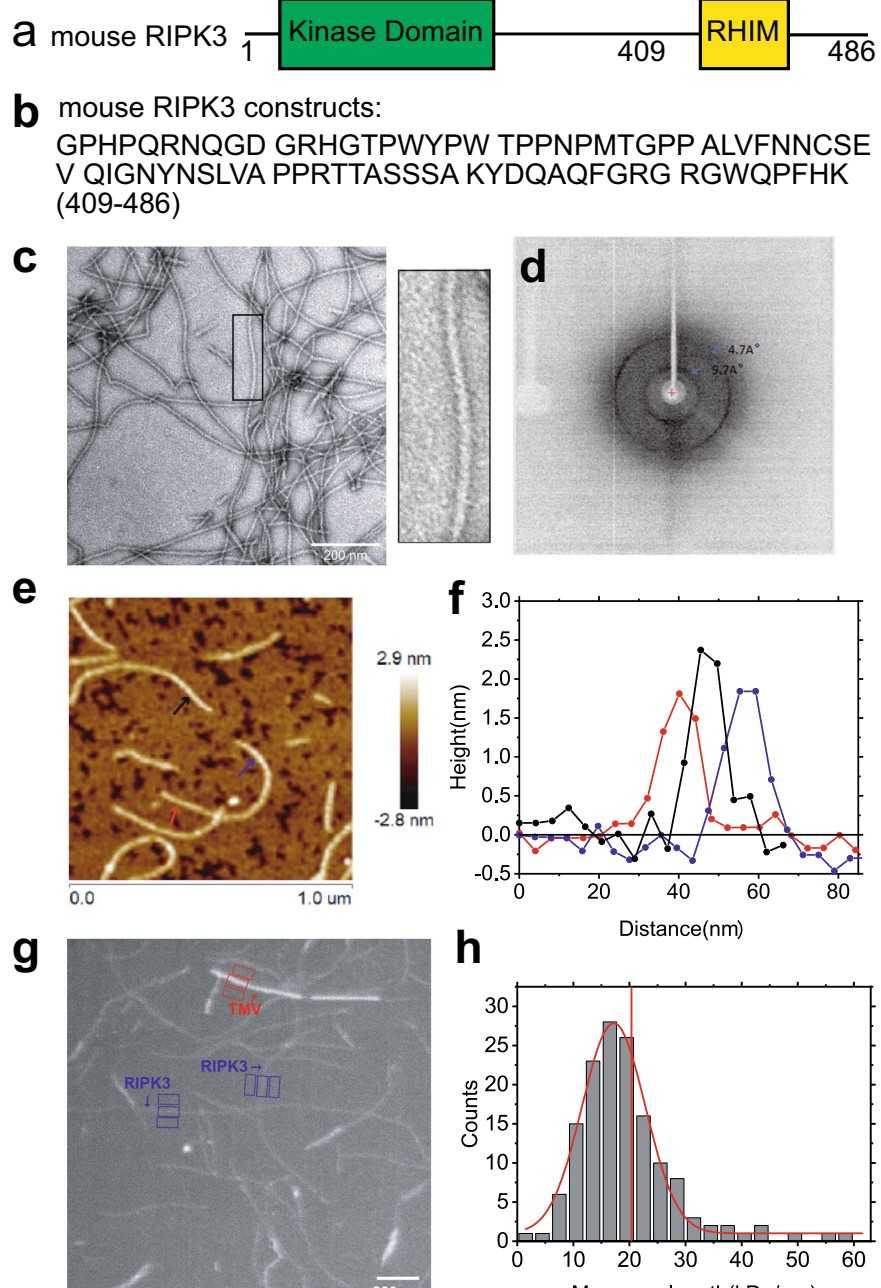

**Fig. 2 The EM, X-ray diffraction, and AFM images of mouse RIPK3 fibrils. a** The domain components of the full-length mouse RIPK3. **b** Protein sequence of mouse RIPK3 construct used for structural elucidation. **c** Electron micrograph of amyloid fibrils (scale bar 200 nm). Mouse RIPK3 fibrils have the straight, unbranched appearance of typical amyloid fibrils. The picture on the right is an expanded view of the boxed area on the left image. The experiment was repeated for more than 5 times. **d** The X-ray diffraction of mouse RIPK3 fibrils. The blue arrow indicated equatorial and meridional reflection at about 9.7 Å and 4.7 Å resolutions, respectively. The experiment was repeated for 3 times. **e** The AFM image of mouse RIPK3 fibrils on mica surface. **f** The height profile at three positions of mouse RIPK3 fibrils corresponding to the three positions indicated by the arrows in (**e**), showing fibril diameter about 1.8 ± 0.2 nm. **g** One BT-TEM image of mouse RIPK3 fibrils, with tobacco mosaic virus (TMV) particles as standards for MPL measurement. **h** MPL histogram of mouse RIPK3 fibrils derived from BT-TEM images. Vertical red line indicated MPL value of 20.3 kDa/nm, the expected value if a single molecule lies in the cross-β unit of the fibril. The variations in the background intensity is analyzed in Supplementary Fig. 2. Figure 1e was a representative image of nine images from three sample preparations and BT-TEM in Fig. 1g was a representative image of more than 30 images from about ten sample preparations.

mixing, only I450 and V residues (V441, V448, and V457) show strong Cα–Cβ cross peaks (Supplementary Fig. 5a), because Cα and Cβ atoms of those residues were simultaneously $^{13}$C-labeled ($^{13}$Cα–$^{13}$Cβ) in each molecule. Other types of residues would have alternating $^{13}$Cα–$^{12}$Cβ or $^{12}$Cα–$^{13}$Cβ labeling patterns because of the properties of [2-$^{13}$C]-glycerol labeling, and thus exhibit no Cα–Cβ cross peaks in $^{13}$C–$^{13}$C spectra with short mixing times. An intermolecular residue $^{13}$Cα–$^{13}$Cβ cross-peak would show up with a longer mixing time if the fibril has in-register parallel β-sheet conformation. We find that with 500 ms mixing (Supplementary Fig. 5b), E447$^{13}$Cα–$^{13}$Cβ, Q449$^{13}$Cα–$^{13}$Cβ, N452$^{13}$Cα–$^{13}$Cβ, N454$^{13}$Cα–$^{13}$Cβ cross peaks show up clearly, indicating in-register parallel intermolecular interactions. Several sequential peaks from residues E447 to V457 are also

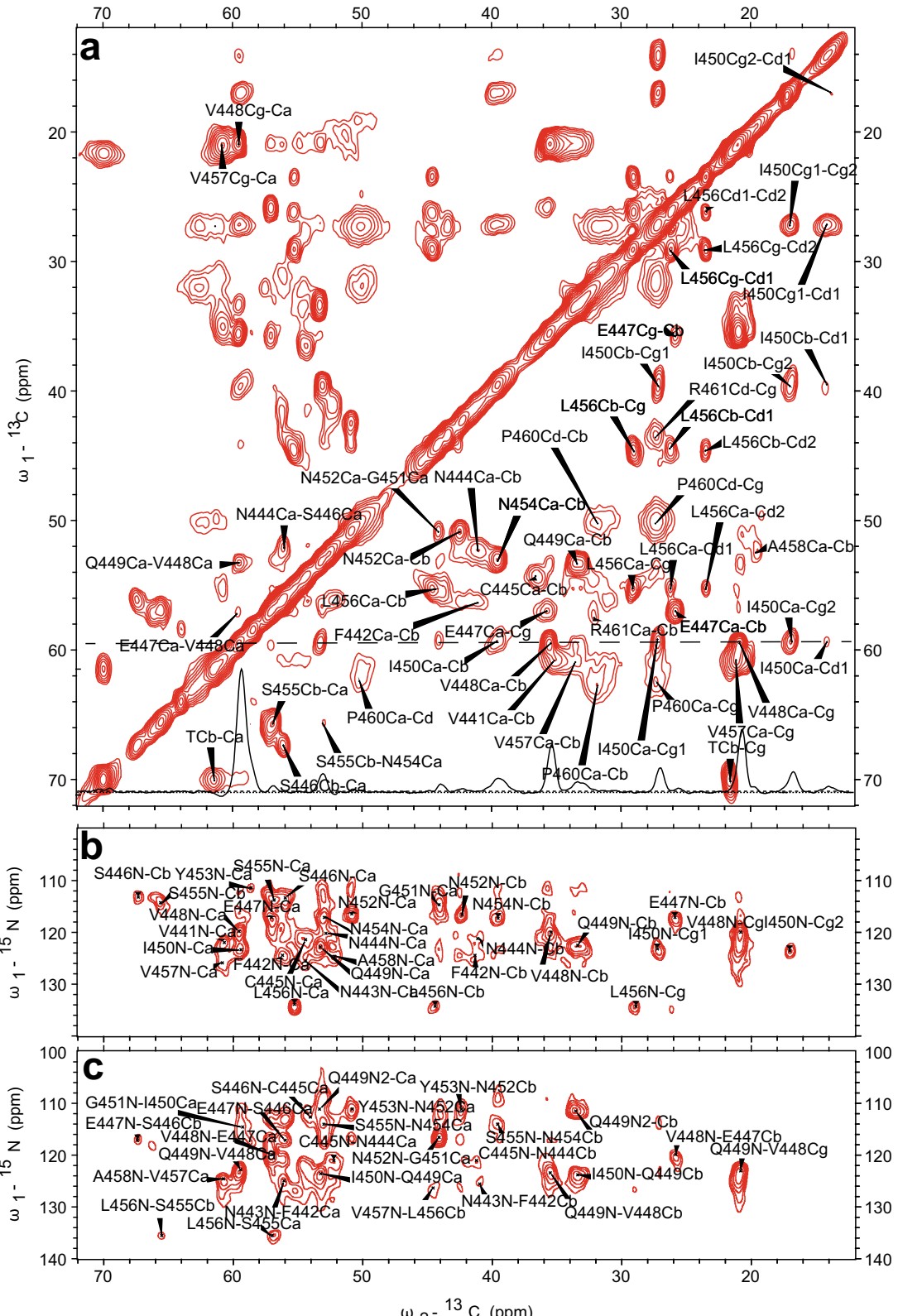

**Fig. 3 SSNMR spectra of uniformly labeled mouse RIPK3 fibrils.** 2D $^{13}C$–$^{13}C$ (**a**), 2D $^{13}C$–$^{15}N$ NcaCX (**b**), and 2D $^{13}C$–$^{15}N$ NcoCX (**c**) with 50 ms DARR mixing. The experiments were carried on a Bruker 700 MHz MAS NMR spectrometer with $\omega_r = 15$ kHz, $T = 303$ K, and 83.33 kHz $^1H$ decoupling field applied during acquisition.

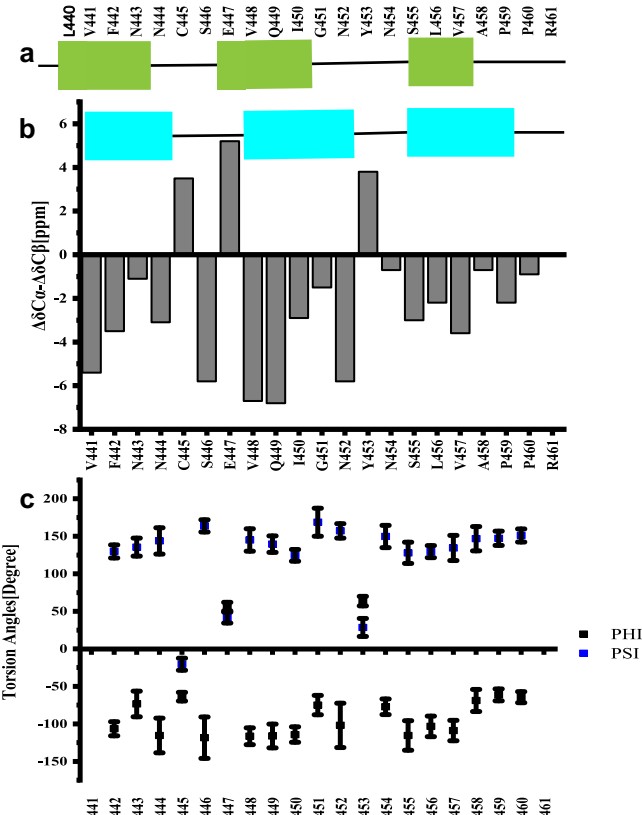

**Fig. 4 Secondary structure prediction from the assigned chemical shifts. a** Secondary structure prediction of mouse RIPK3 construct with PSIPRED[19]. **b** Plot of the difference in the secondary chemical shift between Cα and Cβ, a negative value indicative of β-sheet secondary structures. **c** Predicted protein dihedral angles φ and ψ using TALOS-N based on SSNMR chemical shifts. Data are presented as mean values ± SD from predicted dihedral angles φ and ψ values of 25 best structures.

labeled in Supplementary Fig. 5b, such as V448Cα-E447Cβ, Q449Cα-V448Cβ, I450Cα-G451Cα, Y453Cα-N452Cβ etc. An $^{15}N$–$^{13}C$ z-filtered TEDOR experiment on a fibril sample with half of the molecules $^{15}N$ labeled and the other half $^{13}C$ labeled gives the same conclusion (Supplementary Fig. 5c). By comparing the spectrum to NcaCX spectrum of the uniformly [$^{15}N$, $^{13}C$]-labeled sample, we found the two spectra are well aligned. Most of the residues N-Cα peaks and other peaks, such as V448N-Cβ, Q449N-Cβ, and N452N-Cβ, show up at the same positions for both $^{15}N$– $^{13}C$ correlation spectra. The z-filtered TEDOR spectrum on the mixed sample using two different labels has a lower resolution because it was carried out at a frozen temperature (252 K) for an improved signal/noise ratio. For the same reason, the z-filtered TEDOR spectrum also exhibits more peaks at some positions ($^{15}N$ 130–140 ppm, $^{13}C$ < 20 ppm). The distance between two subunits in the parallel β-sheet conformation is 4.75 ± 0.1 Å, estimated from X-ray diffraction of the fibrils.

The backbone torsion angles ψ and φ and five side-chain torsion angles χ (I450, Y453, N454, L456, and V457) given by TALOS-N predictions from chemical shift values were also used as structural restraints. Xplor-NIH calculations were performed on a fibril represented by 5 copies of residues from 441 to 460. Restraints and the structure statistics are listed in Supplementary Tables 3, 4. The final structure (Fig. 6) was deposited in the Protein Data Bank with PDB ID 6JPD and BMRB entry assigned accession number: 36243.

The calculated mouse RIPK3 fibril structure exhibits three β-strands folding in an "N" shape with the C-terminal β-strand

taking the form of a long and extended tail (Fig. 6a, c). The first and second β-strands adopt the "β-arches" conformation, commonly seen in amyloid fibrils[24]. The RHIM tetrad sequence VQIG in the second β-strand, adopts very ordered side-chain conformations, with V448 and I450 side chains pointing to the first β-strand (Fig. 6b). The side-chain amide groups of Q449 residues are able to form the intermolecular hydrogen bonds with an H…O distance of 1.97 Å, as shown in the fibril structure (Fig. 6d). Aside from Q449, there are three asparagine residues (residue numbers 443, 444, 452) with side-chain amide groups also capable of forming intermolecular side-chain hydrogen bonds (Fig. 6d). The N454 side-chain has an intramolecular contact with G451 (Fig. 6a), pulling the long tail of the 3rd β-strand closer to the first 2 β-strands, but it does not show such hydrogen bonds formation between neighboring molecules here.

**Individual roles of the three β-strands in mouse RIPK3 fibrils.** How important are the three β-strands in determining mouse RIPK3 function? Site-directed mutagenesis of full-length mouse RIPK3 was carried out and the ability of different mutants to induce mouse cell necroptosis was analyzed. TNF-induced necroptosis is mediated by RIPK1, RIPK3, and MLKL. Interestingly, ectopic expression of a functional RIPK3 can convert necroptosis-resistant cells such as mouse NIH-3T3 or human HeLa cells to sensitive ones[4,6,25]. We transfected NIH-3T3 cells (no endogenous RIPK3) with wild-type or mutant forms of mouse RIPK3. Comparing with the wild-type RIPK3, the RIPK3 mutant replacing the 1st β-strand $^{441}$VFNN$^{444}$ or the 2nd β-strand $^{448}$VQIG$^{451}$ to four-alanine residues (AAAA) totally block cell necroptosis (Fig. 7b) while changing the residues in 3rd β-strand of RIPK3 from $^{455}$SLV$^{457}$ to AAA only shows partially inhibition (Supplementary Fig. 6a). Immunoprecipitation assay indicated that the four-alanine mutations of 1st or 2nd β-strand totally inhibited the interaction between mouse RIPK3 and RIPK1 (Fig. 7c), indicating that the intermolecular interaction between mouse RIPK3 and RIPK1 involves more than the conservative RHIM tetrad sequence $^{448}$VQIG$^{451}$. Besides that, single-site RIPK3 mutants F442D, Q449D, and L456D exhibited almost 100% loss in cell necroptosis (Fig. 7b), similar as the whole 1st or 2nd β-strand replacement. The necroptosis function of these RIPK3 mutants were also tested in *Ripk3*-KO MEF (mouse embryo fibroblast) cells and got similar results (Supplementary Fig. 6b). Q449 at the center of RIPK3 RHIM is especially important in stabilizing the fibril structure and determining its function, which has been indicated by previous reports[3]. However, the importance of F442 and L456, two of residues involve intramolecular interactions (Fig. 5), in determining the cell necroptosis indicated in Fig. 7b have never been discussed. Interestingly, more conservative mutations F442A, Q449A, and L456A on mouse RIPK3 only decreased cell necroptosis slightly (Supplementary Fig. 6a), indicating a smaller change in the fibril structure for the conservative mutations. Immunoprecipitation studies on RIPK3 mutants F442D, Q449D, and L456D showed that the intermolecular interaction between mouse RIPK3 and RIPK1 was not affected comparing to wild-type RIPK3 (Fig. 7c), different from the β-strand replacement. Therefore, the change caused by the single-site mutation is not big enough to block the intermolecular interactions in HEK293T. However, the change did inhibit the cell necroptosis.

In order to gain a better understanding on the changes caused by single-site mutations, F442D, Q449D, and L456D mutants with the same construction as the SSNMR sample (mouse RIPK3 409–486) were also prepared to check how the mutation could affect the fibril formation or structure in vitro. The fibril growth

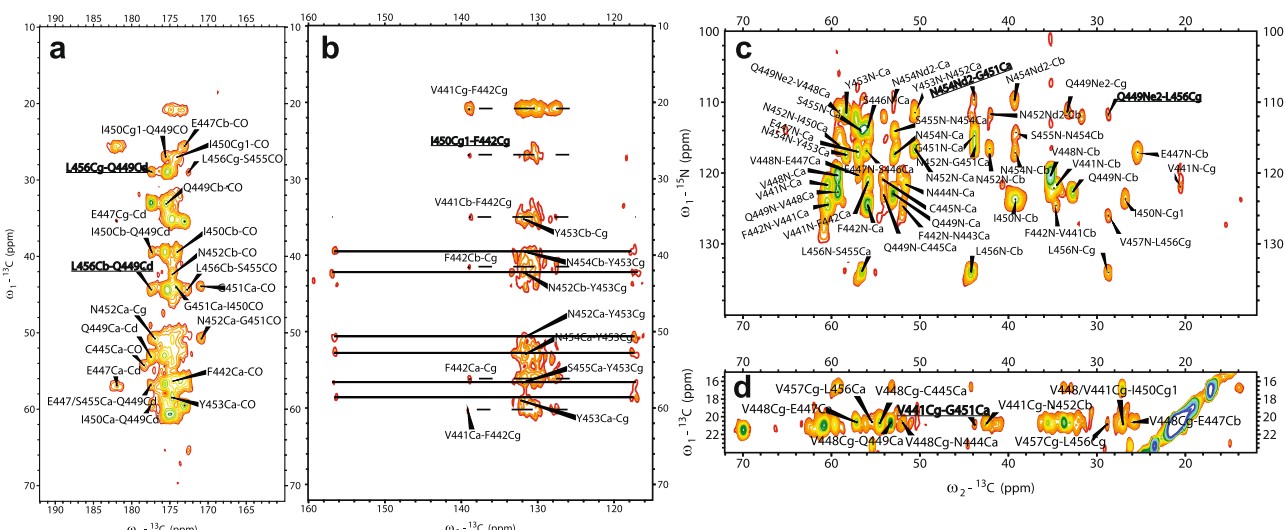

**Fig. 5 SSNMR spectra of uniformly and sparsely ¹³C-labeled mouse RIPK3 fibrils highlighting some long-range inter-residue correlation peaks. a** 2D ¹³C–¹³C correlation spectrum of sparsely ¹³C-labeled mouse RIPK3 fibrils using [2-¹³C]-labeled glycerol with 200 ms DARR mixing, showing the carbonyl region. The long-range correlation peak L456Cβ/Cγ-Q449Cδ are highlighted. **b** 2D ¹³C–¹³C correlation spectrum of uniformly ¹³C-labeled mouse RIPK3 fibrils with 500 ms DARR mixing, showing the aromatic region. The dashed lines indicate the correlation peaks of F442, and the solid lines indicate the correlation peaks of Y453. **c** ¹³C–¹⁵N TEDOR correlation spectrum of sparsely ¹³C-labeled mouse RIPK3 fibrils using [2-¹³C]-labeled glycerol with 6.4 ms z-filtered TEDOR recoupling time. The protein is also uniformly ¹⁵N-labeled. TEDOR shows the correlation peaks of N454Nδ2-G451Cα and Q449Nε2-L456Cγ. **d** 2D ¹³C–¹³C correlation spectrum of sparsely ¹³C-labeled mouse RIPK3 fibrils using [1,3-¹³C]-labeled glycerol with 500 ms DARR mixing. The assignment of V441Cγ-G451Cα is unambiguous.

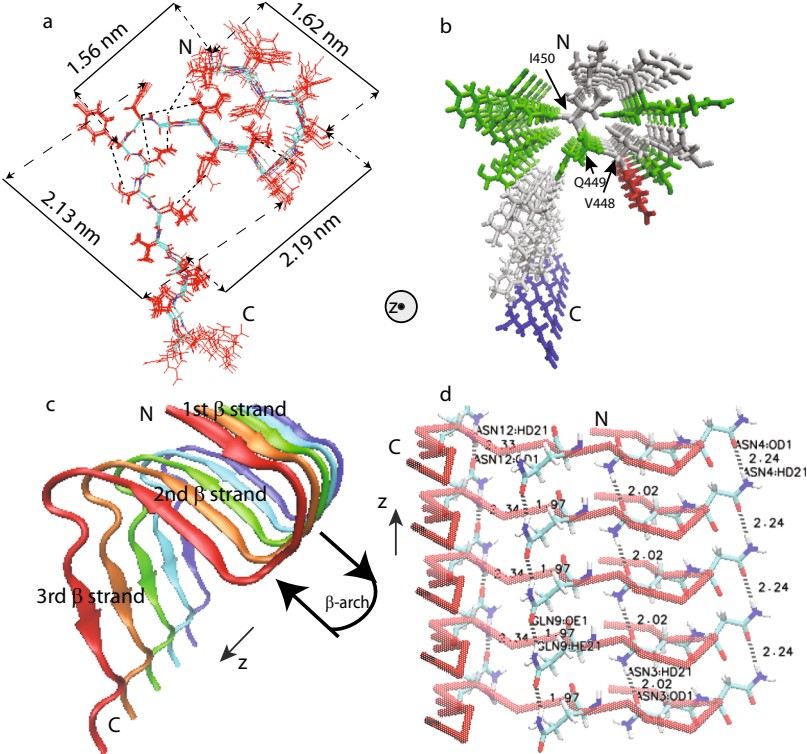

**Fig. 6 Structural model of mouse RIPK3 fibril core (PDB ID 6JPD). a** Superposition of ten monomer conformations with the lowest energy, calculated using XPLOR-NIH software. The dimension of the structure is labeled on the sides. The unambiguous constraints used in the calculation are also marked using the dashed lines. **b** Stick representation of the mouse RIPK3 fibril medoid model selected from 10 fibril structures with the lowest energy. Both (**a**, **b**) are viewed down the fibril axes. **c** Side view of medoid model using cartoon representation. **d** Side view of medoid model indicating possible hydrogen bonding between sides chains of N443, N444, Q449, and N452 (The labels in the figure are ASN3, ASN4, GLN9, and ASN12). All figures were prepared using VMD (https://www.ks.uiuc.edu/Research/vmd/)[49].

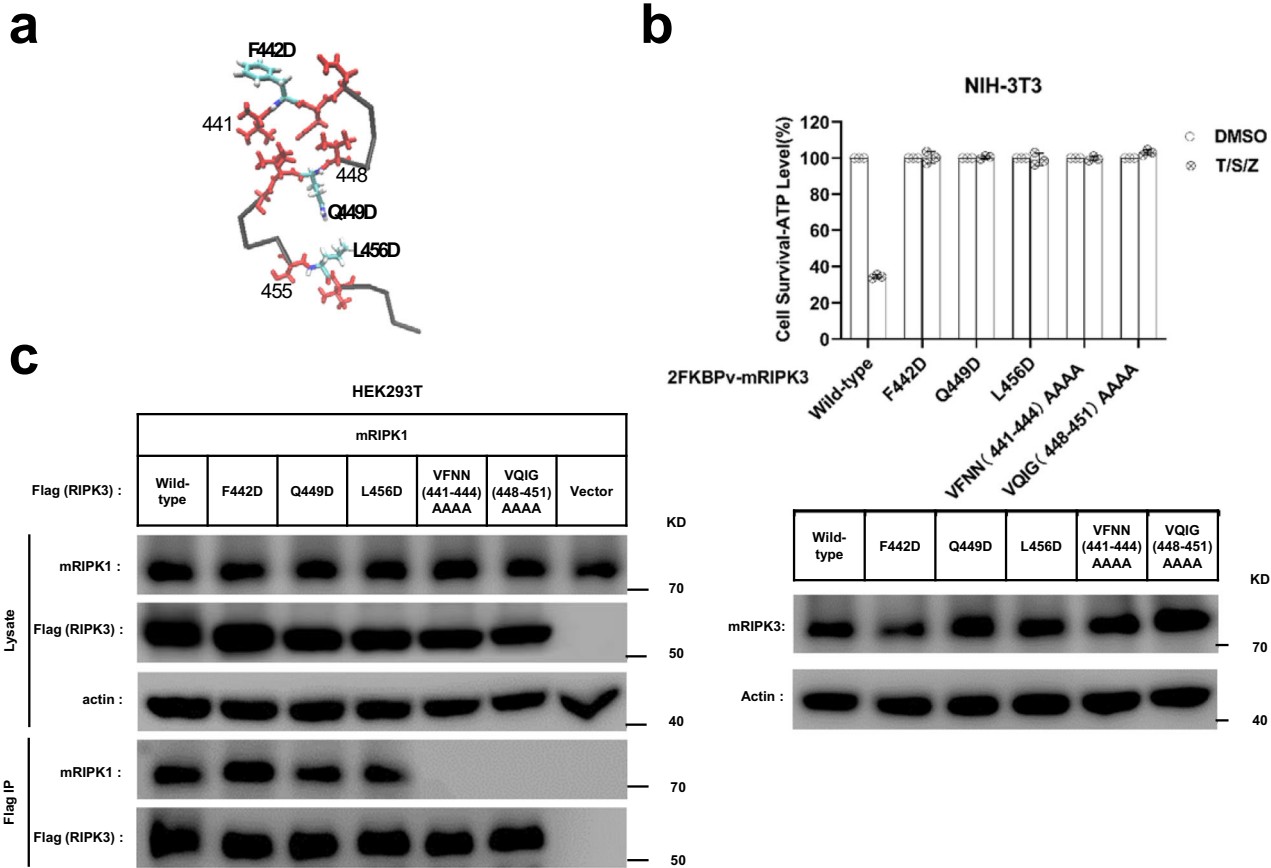

**Fig. 7 Cell-based functional assay. a** The monomer structure of mouse RIPK3, showing the three β-strand segments in red and three single mutation sites. **b** Mutation of F442, Q449, or L456 to D, or quadruple alanine mutations of [441]VFNN[444] or [448]VQIG[451] in RIPK3 led to the complete disruption of the TNF-induced cell necroptosis. The NIH-3T3 cells infected with lentivirus containing FKBPv fused wild-type or mutant RIPK3 were treated with TNF-α/Smac/z-VAD (T/S/Z) 10 h. The number of surviving cells were determined by measuring ATP levels using Cell Titer-Glo kit (upper). Data are presented as mean ± SD of $n = 3$ biologically independent replicates. Source data are provided as a Source Data file. Aliquots of 20 μg whole-cell lysates were subjected to SDS-PAGE followed by western-blot analysis of mouse RIPK3 and β-Actin which was shown as a loading control (lower). **c** The RIPK3 mutant F442D, Q449D, L456D did not affect the interaction between mouse RIPK1 and mouse RIPK3. The HEK293T cells were co-transfected with DNA plasmids containing mouse RIPK1 and Flag-tagged mouse RIPK3 (or its mutants). Cell lysates were collected 36 h post transfection, and immunoprecipitated with anti-Flag magnetic beads (Bimake) at 4 °C. The total cell lysates and immunoprecipitates were analyzed by western-blot analysis with the indicated antibodies. All experiments were repeated three times. The following antibodies were used in this study: anti-mRIPK3 (Sigma-Aldrich, PRS2283, 1:3000); anti-RIPK1 (Cell Signaling Technology, D94C12, 1:1000); Anti-actin (MBL, PM053-7, 1:10,000). Uncropped blots in the Source Data file.

was monitored using THT binding fluorescence (Supplementary Fig. 7a). Q449D and L456D exhibit a gradual increase in the fluorescence intensity for almost 4 h and the maximum intensity is still not observed after the incubation period. On the other hand, wild-type RIPK3 exhibit a rapid increase in fluorescence intensity in the first 20 min and the fluorescence intensity is stabilized after 60 min. F442D shows less fluorescence intensity, but still we could observe a slight increase of fluorescence with time till 60–80 min. The results indicate that mutants have a slower fibril growth rate compared to the wild-type protein. The final fluorescence profiles of fibril upon THT binding were shown in Supplementary Fig. 7b. Q449D shows a small change in the fluorescence intensity while L456D displays a significant increase in the fibril fluorescence intensity. F442D only exhibit little fluorescence, slightly above the blank buffer sample. The change in the fluorescence intensity of fibrils indicate a change of THT binding mode on the fibril, therefore, reflecting the structural changes of the fibrils. Finally, the formed fibrils were visualized using TEM (Supplementary Fig. 7c). TEM images indicate single-strand unbranched fibrils for all. Combined these results, it is clearly concluded that mutants not only have different

fibril growth kinetics, but also have structural changes in the fibrils. SSNMR $^{13}$C–$^{13}$C correlation spectra were also compared for Q449D, L456D, and wild-type RIPK3, giving the structural information at the residue level (Supplementary Fig. 8). The F442Cα–Cβ, S455Cβ–Cα, TCβ–Cα, P460 Cα–Cβ, cross peaks are among the peaks that still remain in the mutant spectra. The peaks for C445, S446, and E447 are disappeared for both mutants suggesting the disordering for the loop connecting 1st and 2nd β-strand. I450 Cα–Cβ shifts to a similar position for both mutants, indicating a structural change at the RHIM core region. N and V residues still could be seen, but the peaks are shifted and the resolution is lost. Our results confirmed that the single-site mutation on RIPK3 (Q449D and L456D) cause changes in the fibril structure.

**The transition process from RIPK1–RIPK3 binding to RIPK3 self-assembly formation.** TNF-induced necroptosis pathway requires RIPK1–RIPK3 intermolecular interaction[3,16]. Meanwhile the functional studies above indicate that the downstream necroptotic process could not be activated without the correct RIPK3 self-assembly formation. In order to build the connection

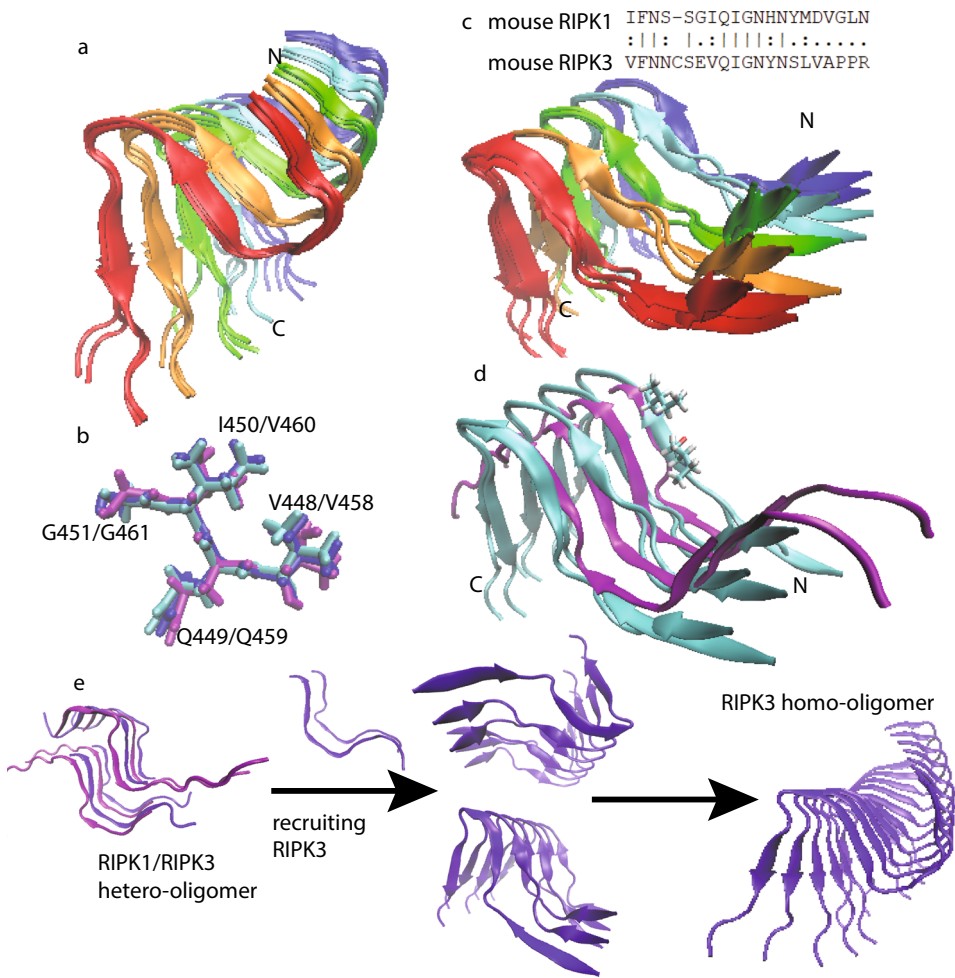

**Fig. 8 Molecular dynamics (MD) simulations using Xplor-NIH on mouse RIPK3 fibrils and a hetero-amyloid model of mouse RIPK1/RIPK3. a** The best four structures of mouse RIPK3 fibril after 50 ps MD, showing the fibril developing a left-hand twist. **b** The structure alignment of RIPK3 conserved tetrad sequence from human RIPK1/RIPK3 hetero-amyloid structure (purple, 5v7z.pdb), mouse RIPK3 fibril structure (blue, 6JPD.pdb) and mouse RIPK3 fibril after 50 ps MD from (**a**) (cyan). **c** The best four structures of mouse RIPK1/RIPK3 hetero-amyloid after 50 ps MD run, showing the opening the β-arches formed by the 1st and 2nd β-strand. The mouse RIPK3 fibril structure was adopted as the starting configuration for the MD. The sequence alignment for mouse RIPK1 and RIPK3 is shown on the top. **d** The structure comparison between **c** (cyan, showing only the best two structures for clarity, residue V448, I450 was also shown in one subunit) and RIPK1/RIPK3 from the human RIPK1/RIPK3 hetero-amyloid structure (purple, 5v7z.pdb). **e** The proposed mechanism showing RIPK3 structural transformation from initial RIPK1–RIPK3 binding to RIPK3 fibril formation.

between the RIPK1–RIPK3 binding and the mouse RIPK3 fibril structure and to understand how the transition would occur, a MD simulation was carried out using the experimental RIPK3 fibril structure as the template. Assuming a 1:1 ratio in the RIPK1–RIPK3 binding, half of the molecules in the fibril structure were replaced by mouse RIPK1. The secondary structure prediction of RIPK1 using PSIPRED shows three sequential β-strand segments with the RHIM tetrad sequence ([528]IQIG[531]) at the 2nd β-strand position (Supplementary Fig. 1b). The positions of the other two β-strands are also spaced similarly to those in mouse RIPK3. This fact strongly suggests that the mouse RIPK1/RIPK3 hetero-amyloid would adopt the parallel in-register conformation seen in the human RIPK1/RIPK3 hetero-amyloid with the tetrad sequence aligned to each other for the different molecules in the fibril. Two MD simulations on the hetero-amyloid model with two possible alignments differing in the 1st loop region were performed to investigate the stability of the RIPK1/RIPK3 hetero-amyloid structure in the configuration of the homo-amyloid structure (Fig. 8c, Supplementary Fig. 9). As a comparison, the same MD simulation was also carried out for a pure mouse RIPK3 fibril.

The final structures of mouse RIPK3 homo-amyloid and RIPK1/RIPK3 hetero-amyloid with the lowest energy after 50 ps MD simulation are shown in Fig. 8a, c respectively. Total 96 calculations were done for each structures. All structures converged after 50 ps and the energy changes during the 50 ps run were shown in Supplementary Fig. 10a. The 96 repeats were done for the statistic purpose. The structural difference within the 96 repeats were shown in Supplementary Fig. 10b comparing the structure with lowest energy and the highest energy. The MD simulation exhibits rather different results for homo-amyloid and hetero-amyloid. The mouse RIPK3 fibril develops a left-hand twist ($6.5° \pm 1°$) without much change in the "N"-shaped 3 β-strands conformation. Interestingly, the N454 side-chain also switches to a conformation forming the hydrogen bonding between the side-chain after the MD run. Additional AFM studies indicated a periodic fibril height changes with a pitch value about 28.5 nm, which gave a twist angle $360/(28.5/0.48) = 6.1°$ (Supplementary Fig. 11). The hetero-amyloid, on the other hand, loses the "β-arches" formed by the first and second β-strand without developing a fibril twists. This result clearly indicates that it is not stable for mouse RIPK1 and RIPK3 to adopt the RIPK3 fibril

structure upon the intermolecular interaction, however, the mouse homo-amyloid structure is stable for itself. Our results suggest a structural transformation may occur for the mouse RIPK3 hetero-amyloid to convert to the final RIPK3 homo-amyloid. A further comparison between the MD relaxation structure of mouse RIPK1/RIPK3 hetero-amyloid and the published structure of human RIPK1/RIPK3 hetero-amyloid shows very similar backbone orientations of the 1st and 2nd β-strand (Fig. 8d), suggesting mouse RIPK1/RIPK3 fibril might be able to adopt a structure similar to that of the human RIPK1/RIPK3 fibrils. Based on this structural evidence, the signal transduction mechanism for the necroptosis signaling during these steps is proposed in Fig. 8e. The hetero-oligomer of RIPK1/RIPK3 is first formed upon TNF induction. While additional RIPK3 molecules are recruited to the hetero-amyloid, RIPK3 conformation transforms and a homo-amyloid of RIPK3 gradually form. The homo-amyloid structure then folds itself to a more compact, stable conformation with 3 β-strands in the molecule.

## Discussion

Our SSNMR structure of mouse RIPK3 fibrils reveal an "N"-shaped structure with three β-strands and a single copy of RIPK3 molecule in the fibril cross-β unit. It shows the β-arch conformation with a strand-turn-strand motif which is a common feature in fibril structures[24]. The β-arch conformation is adopted by the first two β-strands of mouse RIPK3 fibril with residue [445]CSE[447] forming a three-residue β-arc. The β-arches stack on each other, forming two β-sheets which interact with each other via the residue side chains. The third β-sheet adopts an orientation not parallel to the first two, more like a hairpin conformation. This morphology resembles to the published Het-s fibril structure from Podospora anserine (PDB:2KJ3 Fig. 1b), which was also provided by SSNMR[26], Het-s fibril structure is relevant here because it also contains RHIM motif. The RHIM-containing sequence at the C-terminal region of protein HET-s could assemble into highly ordered amyloid fibrils, functioning in a type of programmed cell death, called heterokaryon incompatibility in filamentous fungi[27–29]. Therefore, the SSNMR structure of both RIPK3 and HET-s fibrils provides us high-resolution examples of functional amyloid containing RHIM domains.

In the human RIPK1/RIPK3 hetero-amyloid structure provided by SSNMR, the β-arch conformation is not formed, although the subunits are indeed arranged in a parallel in-register fashion (Fig. 1b). Our MD simulation of mouse RIPK1/RIPK3 structure model favors the human RIPK1/RIPK3 SSNMR structure but not our mouse RIPK3 SSNMR structure, suggesting a more stable conformation for the hetero-amyloid when the first two β-strands adopt an extended orientation (Fig. 8c, d). Moreover, the extended orientation between the first two β-strands exposes the hydrophobic residues (V448 and V450) in the second β-strand to solution (Fig. 8d). In order to maintain a hydrophobic environment for those residues, it would favor another copy of molecules to cover these residues. Therefore, an antiparallel interaction between two RIPK1/RIPK3 protofibrils forms as shown in the SSNMR structure of the human RIPK1/RIPK3 fibrils (Fig. 1b). While in the SSNMR structure of RIPK3 fibrils, V448 and V450 of mouse RIPK3 are buried in the β-arch stabilized by the interactions between the first and the second β-strand where inter-residue cross peaks between F442 and I450 and between V441 and G451 are observed (Figs. 5b, d, 6a). Besides that, the second and third β-strand in mouse RIPK3 fibril forms a hairpin with contacts between the two β-strands shown by inter-residue cross peaks of Q449-L456 and G451-N454 (Figs. 5a, c, 6a). Mutations on those important residues (F442D, Q449D, and L456D) disturb the

stability and interactions between these β-strands, preventing the cell necroptosis (Fig. 7a and Supplementary Fig. 6a, b). It suggests that both β-arch formed by the first two β-strands and the hairpin between the second and the third β-strand are required for RIPK3 function. In the structure of human RIPK1/RIPK3 hetero amyloid, the orientations of the first two β-strands compose a flat turn and the segment corresponding to the third β-strand in RIPK3 fibril is flexible. It suggests the intramolecular interactions shown above between the three β-strands are not needed for RIPK1/RIPK3 hetero amyloid formation. Consistent with that the mutations of human RIPK1 (I533D or M547D) or RIPK3 (I452D or L466D) corresponding to mouse RIPK3 mutations (F442D or L456D) did not disrupt the RIPK1/RIPK3 hetero fibrillar complex in vitro[3]. Our Immunoprecipitation results also showed that F442D, Q449D and L456D did not affect the interaction between RIPK1 and RIPK3 while these mouse RIPK3 mutations did disrupt their necrosis function. Only the 4-residue segmental replacement into AAAA at [441]VFNN[444] (in the first β-strand) or [448]VQIG[451] (in the second β-strand) was significant to disrupt the RIPK1–RIPK3 binding (Fig. 6a, b). These pieces of evidence strongly suggest that the formation of hetero fibrils composed by RIPK1 and RIPK3 is necessary but not sufficient for RIPK3 dependent necroptosis. RIPK3 must transit to form the unique "N"-shaped fibrils to transduce the necroptosis signal.

How does necroptosis signal transit from upstream RHIM-containing factors to RIPK3? Comparing the fibril structures of human RIPK1/RIPK3 and mouse RIPK3, we found the conformation of the 4-conserved residues I(V)QI(V/L)G in the second β-strand are nearly identical (Fig. 8b). It suggests the tetrad sequence of RIPK1 or other upstream RHIM-containing factors will fold as amyloidal nucleates to recruit the second β-strand of RIPK3 first. Consistently, mutations to replace the 4-conserved residues of either upstream factors including RIPK1 and TRIF or RIPK3 (Fig. 7a) will prevent the RIPK3 recruitment and cell necroptosis[3,30,31]. Then, the second β-strand of RIPK3 RHIM behaves as the amyloid core, to induce the stacking of the first and the third β-strand to form a unique "N"-shaped structure (Fig. 8e). The RIPK3 mutations (F442D, Q449D, and L456D) that destabilize the fibril or delay the fibril formation would prevent the cell necroptosis.

## Methods

**Production of mouse RIPK3 protein**. All RIPK3 constructs were subcloned into pSMT3 vector with a N-terminal 6 × His tag. For mutant proteins (F442D, F442A, Q449D, Q449A, L456D, and L456A), a QuikChange protocol was used to obtain the mutant protein sequence (Sangon Biotech, Shanghai). The required primers were designed according to the protocol provided in the QuikChange Manual (Supplementary Table 5). Polymerase Kod-201(Toyobo, Shanghai), an enzyme with high fidelity, was used to amplify the plasmid with the primers so that the single-site mutant plasmid could be obtained through PCR. PCR product was digested with restriction enzymes DpnI(NEB, USA) for 1 h at 37 °C and then purified using DNA purification kit (Takara). The purified plasmid was then transformed into DH5α competent cells (Transgene, China). The mutations were all confirmed by the sequencing.

All proteins were expressed in *E. coli* Transetta (DE3) cells (Transgene, China). Unlabeled protein was expressed in 1 L Luria broth medium (Sangon Biotech, Shanghai) supplemented with 100 mg/mL ampicillin (Sangon Biotech, Shanghai) and the cells were induced for expression at an $OD_{600} = 0.8$–1.0 determined by a BIOMATE 3S UV–Visible Spectrophotometer (Thermo Fisher, USA). After 4 h of induction at 37 °C with 0.8 mM Isopropyl β-D-Thiogalactoside (Sangon Biotech, Shanghai), cells were harvested by centrifugation at $13,881 \times g$ for 10 min and lysed by high pressure nano homogenizer (FB-110X, Shanghai Litu Ins., China) at about 850 bar in a buffer containing 50 mM Tris-HCl (pH 8.0), 300 mM NaCl, 2 mM phenylmethylsulfonyl fluoride (BBI Life Science), and 1 mM β-mercaptoethanol (Sigma). After centrifugation at $24,330 \times g$ for 30 min, the pellet was dissolved in 20 mL dissolving buffer containing 6 M guanidine hydrochloride (GudHCl) (General-Reagent, Shanghai Titan Scientific), 50 mM Tris-HCl (pH8.0), and 300 mM NaCl. The mixture was again centrifuged at $24,330 \times g$ for 30 min and the supernatant was incubated with Ni-NTA beads 6 FF (Smart-Life Science) at 4 °C for 30 min. The protein was later eluted from the beads using 15–20 mL dissolving buffer containing 250 mM imidazole (Sangon Biotech, Shanghai). Then the protein solution was dialyzed using 1 L pure water at 4 °C with Spectra/por@6 Dialysis

Membranes (MWCO 3.5 K, W.45 mm, Diam 29 mm; BBI Life Science) and water was replaced twice after every 6 h. The protein precipitation was harvested by centrifugation at 24330×g for 10 min and dissolved again using 25% (v/v) acetic acid solution for further purification. The protein was finally purified by reverse phase high-performance liquid chromatography (Waters 2545) at room temperature with a linear gradient of 30–70% aqueous-organic solvent over 10 min at 10 mL/min using the XBridge@ Peptide BEH C18 column (130 Å pore, 5.0 μm beads, 19 mm × 100 mm column, Waters). Aqueous phase was Milli-Q $H_2O$ with 0.05% TFA and the organic phase is acetonitrile with 0.05% TFA. The purified protein was flash-frozen in liquid nitrogen and dried at −80 °C. Protein concentrations were determined by absorbance at wavelength of 280 nm.

The uniformly labeled protein, [$^{13}$C, $^{15}$N]-labeled mouse RIPK3, was expressed in the freshly prepared M9 medium containing 1.5 g/L $^{15}$N-ammonium chloride and 2 g/L $^{13}$C-glucose (Cambridge Isotope Laboratories), 1 mL/L BME vitamins (Sigma-Aldrich B6891), 0.2 M $CaCl_2$, 2 M $MgCl_2$, and 50 mg/L Thiamine. The culture grew first in 500 mL LB medium with shaking (220 rpm) at 37 °C to the cell density of about $OD_{600}$ = 1.0. Cells were then collected by centrifugation at 4 °C and resuspended in the freshly made M9 medium. After the cells' recovery for 30 min at 37 °C, the protein expression was induced using 0.8 mM isopropyl β-d-1-thiogalactopyranoside for 4 h at 37 °C with shaking at 220 rpm. The protein was purified with the same method described above.

For [$^{12}$C, $^{15}$N]-labeled protein, the expression medium use 1.5 g/L $^{15}NH_4Cl$ and 2 g/L $^{12}$C-glucose as the nitrogen and carbon source; For [$^{13}$C;$^{14}$N]-labeled protein, the expression medium use 1.5 g/L $^{14}NH_4Cl$ and 2 g/L $^{13}$C-glucose as the nitrogen and carbon source. And for sparsely $^{13}$C-, uniformly $^{15}$N-labeled proteins, [2-$^{13}$C]-glycerol or [1,3-$^{13}$C]-glycerol was used as the carbon source.

**Fibril sample preparation.** Lyophilized protein powder was first dissolved in 6M GudHCl solution (pH 7.5) at a concentration of 1 mg/mL. The protein solution was incubated for 1 h to make sure the fully dissolving of the protein. It was then overnight dialyzed in Milli-Q water at room temperature with Spectra/por@6 Dialysis Membranes (MWCO 3.5 K, W.45 mm, Diam 29 mm; BBI Life Science). The pH of Milli-Q water used was adjusted to 7.5 using 1M NaOH buffer. Water was then changed every 6 h for additional three times. During the process, protein fibrils formed gradually. After keeping the sample in the dialysis membrane for 3 days, protein pellets were then collected by ultracentrifugation at 25,9000 × g, 25 °C for 1 h. (Optima Max-TL, BECKMAN COULTER). For protein fibril preparation using [$^{12}$C, $^{15}$N]- and [$^{13}$C, $^{14}$N]-labeled protein, [$^{12}$C, $^{15}$N]-labeled protein and [$^{13}$C; $^{14}$N]-labeled protein were prepared separately and mixed in 1:1 mole ratio before further dialysis to remove GudHCl.

**X-Ray diffraction (XRD) from fibrils.** The fibril pellet was mounted in a loop and exposed to Cu $κα$ radiation from a Bruker D8 VENTURE X-ray diffractometer at 0.154184 nm wavelength, distance 50 mm. Data were collected at room temperature for 1 min on a Bruker D8 VENTURE imaging plate detector by APEX3 V2018.7-2 software. The XRD images were processed with the adxv.x86_64RHEL6 program (Scripps Research institute, LaJ olla, CA, USA).

**Thioflavin T fluorescence binding assays.** ThT binding assay was performed to monitor the kinetics of fibril growth for mouse RIPKs and its mutant using a Perkin-Elmer EnSight Multimode Plate Reader with a Costar 96-well plate (Corning). The excitation wavelength was at 430 nm and the emission was monitored at 485 nm. ThT was at a concentration of 50 μM and protein fibrilization was carried out by diluting 2 mM protein stock solution in 6 M GudHCl to a final concentration of 20 μM in a 200 μL volume using 10 mM PB buffer (pH 7.4). The data were collected by measuring fluorescence intensity continuously for about 4 h at room temperature and were plotted using Origin2018. The fluorescence profile of RIPK3 fibrils from 450 to 600 nm was also obtained.

**Electron microscopy.** Five microliters of the fibril suspension was dropped onto a 300-mesh carbon-coated grid (Beijing Zhongjingkeyi Technology). The solution was kept for 5 min on the grid before wicked off by filter paper. The grid was washed twice by 5 μL Milli-Q water and stained with 5 μL 2% uranyl acetate in water (w/v) for negative staining. Then the excess of liquid was blotted off and the grid was allowed to air dry. TEM images were recorded using Tecnai G2 Spirit Transmission Electron Microscope operating at 120 keV.

**Mass-per-length measurement of mouse RIPK3 fibril using beam tilted (BT)-TEM.** Mouse RIPK3 fibrils mixed with diluted TMV (generously provided by the laboratory of Jun Yang at Wuhan Institute of Physics and Mathematics, Chinese Academy of Sciences) were adsorbed onto a 200-mesh carbon-coated copper grid with the carbon film 3–5 nm in thickness. Images were acquired by a Talos L120C TEM at an acceleration voltage of 120 kV. BT-TEM images were taken at ×36,000 magnification, using a beam tilt of 1.2°, a 70 μm diameter objective aperture, a 150 μm diameter condenser aperture and a spot 2 setting with a filament current of about 5 μA. The dose rate was 15–20 e/nm$^2$ s when taking the images which were later stored as 8-bit tiff files. Images were analyzed with ImageJ (NIH, 1.52 v) and MPL values were calculated based on the reference[22]. We obtained 147 MPL counts from 6

dark-field images with each rectangle size of 60 nm × 120 nm. MPL error analysis were also calculated according to the method in the reference with 139 counts.

**Atomic force microscope.** Five microliters of 50 times diluted fibril solution was deposited onto the freshly cleaved mica surface and incubated for 5 min at room temperature. The mica sheets were subsequently rinsed twice with 10 μL Milli-Q water to remove the unbound material and air dried. The imaging for fibril height measurements was performed under dry conditions in a tapping mode using 0.01–0.025 Ω cm n-type Antimony(n) doped Si cantilevers (model RTESPA-300, Bruker, US) at about 300 kHz on a Dimension Icon AFM with Bruker Nano scope V controller (Digital Instruments, Goleta, CA, USA). The images were recorded at a scan rate of 1 Hz, acquiring 256 points per line and 256 lines over a 1 μm$^2$ area. Fibril diameters were estimated using the fibril height measured from the AFM images subtracting the average baseline in a 1 μm section across the fibril. The final value is the result of 21 measurements on eight different fibrils in four different images. The imaging for the later fibril twist measurements was recorded with a PeakForce tapping mode using a SCANASYST-AIR-HR probe (Bruker, US) on Multimode 8. The images were recorded at a scan rate of 0.814 Hz, acquiring at 512 × 512 pixels over a 1.2 μm$^2$ area.

**Solid-state NMR experiments.** Most of the magic angle spinning (MAS) SSNMR experiments were carried out on a 16.45 T (700 MHz $^1$H frequency) Bruker AVANCE NEO spectrometer. A 3.2 mm triple-resonance HCN MAS probe was used. All experiments were conducted at 303 K with MAS rate ($ω_r$) of 15 kHz. $^{13}$C chemical shifts were externally referenced to DSS using the published shift of adamantine (40.48 ppm for downfield $^{13}$C signal). And $^{15}$N chemical shifts were referenced to liquid ammonia (0.00 ppm of $NH_3$) using the IUPAC relative frequency ratios between DSS ($^{13}$C) and liquid ammonia ($^{15}$N). All spectra were processed using topspin and analyzed using the program Sparky[32]. Some NMR experiments were performed on Agilent 700 NMR equipment and the spectra were processed with nmrPipe software.

For $^{13}$C–$^{13}$C 2D correlation experiments, the Hartman–Hahn cross polarizations (CP) were done with a $^{13}$C field strength of 51.2 kHz and the $^1$H field strength adjusted to near the $n = 1$ Hartman–Hahn condition, 68.2 kHz. The CP contact time was 1.5 ms. The $^1$H and $^{13}$C hard pulse radio frequency (rf) field strengths were 83.3 kHz and 75.7 kHz, respectively. Dipolar-assisted rotational resonance (DARR)[33] was applied for $^{13}$C–$^{13}$C polarization transfer with mixing time of 50, 200, and 500 ms. SPINAL-64 decoupling[34] was employed during t1 and t2 increment with a $^1$H rf-field strength of 83.3 kHz.

For 2D NcaCX and NcoCX, the $^1$H–$^{15}$N CP was done using the $^{15}$N field strength of 50 kHz and the $^1$H field strength adjusted to near the $n = 1$ Hartman–Hahn condition. For the SPECIFIC CP transfer[35], a mixing time of 4.5 ms was used with rf-field strengths about 2.7 $ω_r$ ($^{15}$N) and 1.7 $ω_r$ ($^{13}$C) for NCA, and 1.7 $ω_r$ ($^{15}$N) and 2.7 $ω_r$ ($^{13}$C) for NCO, respectively. A DARR sequence of 50 ms mixing time was used for subsequent $^{13}$C–$^{13}$C polarization transfer. During acquisitions, a TPPM $^1$H decoupling scheme with rf field of 87.72 kHz was applied[36].

For z-filtered transferred echo double resonance (z-filtered TEDOR) experiment[37], the $^{13}$C and $^{15}$N hard pulse rf field strengths were 75.7 and 50 kHz, respectively. During the magnetization transfer, $^{15}$N π pulse length was phase cycled according to the xy-4 scheme. The z-filtered time was 200 μs. $^{13}$C–$^{15}$N TEDOR mixing time was set to 3.2, 6.4, and 8.5 ms, respectively.

2D $^1$H–$^{13}$C insensitive nuclei enhancement by polarization transfer (INEPT)[38] was carried out for detecting the mobile part of the fibrils. The J-coupling value was set to 140 Hz for the general $^1$H–$^{13}$C transfer. The waltz decoupling of 20 kHz was employed on $^1$H channel during acquisition. INEPT–$^{13}$C–$^{13}$C-total through-bond-correlation spectroscopy (INEPT-TOBSY) experiment[39,40] was carried out with a TOBSY mixing time of 11.2 ms using the P9$^1_6$ mixing sequence.

**Calculation of structural models for mouse RIPK3 fibrils.** Structure calculations were performed using simulated annealing with the Xplor-NIH package[41]. Two rounds of calculations were carried out. Mouse RIPK3 molecule residues V441-P460 were used in the calculation since those residues had chemical shifts assignments and TALOS-N predictions of protein dihedral angle values. In these calculations each RIPK3 subunit is assumed to have the exactly same conformation. To enforce this condition, the strict symmetry module (symSimulation) in the Xplor-NIH package (2.48) was utilized to reduce the computational cost, where only a single copy of protomer coordinates were maintained[42]. In total, 5 copies of the monomer subunit were used in the calculation to represent a short fibril segment, where 4 subunit copies were generated from a protomer using rigid body translations; nonzero twist angle was not considered.

In the first round of calculation, 108 independent structures were calculated from starting coordinates having different, random torsion angles and packing. The protocol contains first torsion-angle dynamics for a duration of 10 ps or 5000 timesteps at 4000 K, followed by annealing to 25 K in decrements of 12.5 K for 20 ps or 2000 timesteps of torsion-angle dynamics at each temperature and finally 500 steps of energy minimizations in torsion angle and Cartesian coordinates. The calculation was done on the high- performance calculation platform of ShanghaiTech University. Based on MPL data, there is only one protofibril in a mature fibril structure. Backbone torsion angles (using the CDIH potential) were restrained using predictions from both TALOS-N[23]

and TALOS+[43]. Although the TALOS-N predicted values were used for the structure calculations, only those predictions whose values agreed within 20 degrees were used and the uncertainties were expanded to accommodate the differences between the two methods. Intermolecular distance restraints (using the NOEPot potentials) were applied between neighboring subunits for the N444Cβ, V448Cβ, and S455CO atoms, using a carbon-carbon distance of 4.75 ± 0.1 Å, and explicitly representing intermolecular hydrogen bonds between N444NH and N443CO, C445NH and N444CO, Q449NH and V448CO, I451NH and G450CO and L456NH and S455CO, using hydrogen-oxygen distances of 2.3 ± 0.1 Å and nitrogen-oxygen distances of 3.3 ± 0.1 Å. These intermolecular bonds were employed so that the resulting fibrils are consistent with the 4.7 Å peak seen in X-ray powder diffraction. Intramolecular long-range distance restraints were obtained from 2D $^{13}$C–$^{13}$C correlation using 200 ms or 500 ms DARR mixing (distance restraint values: 5.5 ± 1.5 Å) and z-filtered TEDOR with $^{13}$C -$^{15}$N recoupling time 6.4 ms (distance restraint values: 4.5 ± 2.5 Å), and comprised ten unambiguous restraints between the pairs V441Cγ-G451Cα, S446Cβ-V448Cβ, Q449Cδ-L456Cβ/Cγ, I450Cγ2-N452Cα, G451Cα-N454Nδ2, N452Cα/Cβ-N454Cα, Y453Cβ-S455Cβ, and L456Cγ-Q449Nε2. Long-range distance restraints with low ambiguity from DARR and TEDOR were also used in the first-round calculation, usually with one site having a unique assignment and the other site having two possible assignments. Aside from the experimentally-based dihedral and distance restraint terms, the knowledge-based TorsionDB[44], low-resolution residueAff[45] contact terms, along with the standard purely repulsive nonbonded RepelPot (Schwieters et al.[42]) and covalent bond, bond-angle and improper dihedral terms were used in this initial docking calculation. The best five structural models with the lowest energy were retained for the second round of structure calculation.

The second refinement round of calculation was similar to the initial folding calculation except that the RepelPot term was replaced by the EEFx[46] implicit solvent force field. 5 side-chain χ values from TALOS-N predictions were also added into the CDIH potential term to improve the side-chain conformation. Long-range distance restraints with high ambiguity from DARR and TEDOR were introduced where neither sites had unique assignments. The values used in the distance restraints with low ambiguity and high ambiguity in the refinement were set to be 5.5 ± 2.5 Å. A total of 200 structures were calculated and the best 10 structures with the lowest energy were validated at https://validate-rcsb-1.wwpdb.org/[47]. Structural statistics are shown in Supplementary Table 3. The mouse RIPK3 fibril structural models were deposited into the Protein Data Bank with the PDB ID:6JPD and NMR chemical shifts were deposited into BMRB: 36243.

**Molecular dynamic (MD) simulations**. The docking of mouse RIPK1 into the RIPK3 fibril was also carried out with Xplor-NIH, using the RIPK3 fibril medoid model as the template. A mouse RIPK3 fibril with 10 subunits was first generated and every other one subunit then replaced by a mouse RIPK1 molecule. The RIPK3 and RIPK1 molecules were aligned at the tetrad sequence for the 2nd β-strand, however, two alignments would still be possible for RIPK1 and RIPK3 at the 1st β-strand (Fig. 7c and Supplementary Fig. 8). The docking for both alignments were carried out for comparison. An all-atom energy minimization of 1 ps was then carried out in which the positions of the backbone atoms of the two RIPK3 subunits were restrained to remain within 1 Å of their initial positions using PosDiffPot (Schwieters et al.[42]). In the energy minimization, the XplorPot, TorsionDB, implicit solvent (EEFxPot)[48], and covalent energy terms were also included. Minimization was followed by 1 ps of MD with randomized initial velocities appropriate to 300 K for initial equilibration. After this, the PosDiffPot was disabled and MD was performed for 50 ps at 300 K. A total of 96 runs were performed for each of the mouse RIPK3 and the two RIPK1/RIPK3 docking models of the fibril. The best four structures were shown for comparison.

**Constructs and transfection**. For lentivirus production, the wild-type and mutated RIPK3 cDNAs were cloned into the modified lentiviral vector pCDH-CMV-MCS-EF1-copRFP. HEK293T cells (ATCC) were seeded on 10 cm dishes and cultured to 70% confluence. The cells then were transfected with the prepared lentiviral vectors and virus packing plasmids (psPAX2 and pMD2.g, Addgene) by using EZ transfection reagents (Shanghai Life-iLab Biotech Co., Ltd). The virus-containing medium was harvested 48 h later and added to the NIH-3T3 (ATCC) or Ripk3-knockout MEF cells were generated from Ripk3-knockout genotype of mice as indicated with 10 μg/ml polybrene. The infection medium was changed with fresh medium 24 h later. Cells with stable expressed RIPK3 were selected at 72 h post infection by FACS.

**Cell survival assay**. The sources of the materials used in this study: recombinant TNF-α was purified in Hua-yi's lab, Smac mimetic was kindly provided by Xiao-dong Wang (NIBS, China). and z-VAD (cat# 1140) was bought from BioVision. NIH-3T3 or Ripk3-knockout MEF cells with wild-type or mutant RIPK3 expression were cultured to 90% confluence, then they were digested and seeded in 96-well plates. Approximately 6000 cells were seeded in each well, including two duplicate wells. After 12 h, necroptosis was induced by adding the final concentrations of 10 ng/ml TNF-α (T), 100 nM Smac mimetic (S), and 20 μM z-VAD (Z) to the cell culture wells. After 10 h, cell survival was determined by measure cellular ATP level with the Cell Titer-Glo Luminescent Cell Viability Assay kit. A Cell Titer-Glo Luminescent Cell Viability Assay (Promega) was performed

according to the manufacturer's instructions. Luminescence was recorded with an EnSpire Multimode Plate Reader from Perkin-Elmer.

**Immunoprecipitation and immunoblotting**. HEK293T cells were cultured on 10 cm dishes and grown to 75% confluence, then transfected with DNA plasmids containing mouse RIPK1 and Flag-tagged wild-type or mutant mouse RIPK3 using EZ transfection (Shanghai Life-iLab Biotech Co., Ltd.). Thirty-six hours later, cells treated as indicated were washed once with DPBS and harvested by scraping and centrifugation at 1000 × g for 3 min The harvested cells were washed once with DPBS and lysed for 30 min on ice in lysis buffer containing 25 mM Hepes-NaOH (pH 7.5), 150 mM NaCl, 1% Triton, 10% glycerol, and complete protease inhibitor (Roche) and phosphatase inhibitor (Sigma) cocktails. The cell lysates were then centrifuged at a top speed of 12,000 × g for 30 min at 4 °C. The soluble fraction was collected, and the protein concentration was determined by a Bradford assay. For immunoprecipitation, 1 mg of extracted protein in lysis buffer was immunoprecipitated overnight with anti-Flag magnetic beads (Bimake) at 4 °C. After incubation, the beads were washed three times with lysis buffer, then directly boiled in 1X SDS loading buffer and subjected to immunoblot analysis. The following antibodies were used in this study: anti-mRIPK3 (Sigma-Aldrich, PRS2283, 1:3000); anti-RIPK1 (Cell Signaling Technology, D94C12, 1:1000); Anti-actin(MBL, PM053-7, 1:10,000). For protein expression analysis in cell survival experiments, the samples were subjected to SDS-PAGE and detected using antibodies as indicated. Uncropped blots in the Source Data file.

**Reporting summary**. Further information on research design is available in the Nature Research Reporting Summary linked to this article.

## Data availability
The SSNMR structure of mouse RIPK3 was deposited as PDB ID: 6JPD and NMR chemical shifts were deposited into BMRB: 36243. All NMR data were available from the corresponding authors on reasonable request. Source data are provided with this paper.

## Code availability
All the structural calculation scripts were available from the corresponding authors on reasonable request.

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

## Acknowledgements

The work is partially supported by grants from National Key R&D program of Ministry of Science and Technology of China (2017YFA0504804 to J.X.L.), National Natural Science Foundation of China (No. 31770790 to J.X.L. and 31571427 to H.Y.W.). This work is also funded by the scientific research start-up of J.X.L. from the school of life science, Shanghaitech University. J.W. is sponsored by Yangfan program of Shanghai municipal government (19YF1433500). The authors thank Dr. Yan-fen Liu for gifting the plasmid. We thank Dr. Xin-yan Wang and Na Yu at the Analytical Instrumentation Center of the school of Physical Science and Technology (contract no. SPST-AIC10112914) at ShanghaiTech University for the assistance in the AFM and XRD experiments. Additional AFM studies were carried out with the help from Dr. Liu Cong's lab at Interdisciplinary Research Center on Biology and Chemistry, Shanghai Institute of Organic Chemistry, Chinese Academy of Sciences. We thank Dr. Hou-chao Tao for the aid in the HPLC instrument. Our TEM and NMR work were partially performed at National Center for Protein Science Shanghai, China and at the Bio-Electron Microscopy Facility, the Biomolecular NMR Facility of the School of Life Science, Shanghaitech University. We thank Dr. Bin Wu and Hong-juan Xue from National Center for Protein Science Shanghai, China for the help on the NMR. We thank Dr. Qian-qian Sun and Dr. Dan-dan Liu from the facility of School of Life Science, Shanghaitech University for the assistance on TEM. TMV was provided by the laboratory of Jun Yang at Wuhan Institute of Physics and Mathematics, Chinese Academy of Sciences. C.D.S. is supported by the Intramural Research Program of the Center for Information Technology of the National Institutes of Health. The structural calculations were partially done using the high-performance calculation platform of Shanghaitech University.

## Author contributions

This paper is completed under the efforts of all authors. X.W. prepared all samples for the structural studies, took AFM images, captured the EM and BT-TEM dark-field images and SSNMR experiments; H.H. carried out the functional studies in cells; X.D. helped in the EM images and the structural calculations; J.Z. collected the fiber diffraction data; J.W. helped in SSNMR experiments; J.X.L., H.W., X.W., and H.H. planned experiments and did data analysis; J.L., G.W., J.Y.L., and B.L. helped to purify the protein and give suggestions to data analysis. J.X.L., X.D., and C.S. carried out the structure calculations; X.W., H.W., and J.X.L. wrote the paper.

## Competing interests

The authors declare no competing interests.
