## [Peer Review File · Nature Communications]

REVIEWER COMMENTS

Reviewer #1 (Remarks to the Author):

Lu and co-workers investigate the role of mouse RIPK3 in cell necroptosis. This study includes a functional investigation of the roles of RIPK1 and RIPK3, a solid-state NMR based structural study of mouse RIPK3 fibrils, and an MD study to establish the stability of RIPK3 fibrils, but the instability of the RIPK1/RIPK3 hetero-amyloid when forced to take the same structure as the RIPK3 fibrils. The hetero-amyloid adopts a structure similar to a previously determined structure of human RIPK1/RIPK3 hetero-amyloids.

The RIPK3 structure, established based on solid-state NMR and MPL measurements, is well-supported. Functional studies establish the relative importance of the β -strands found in RIPK3 in inducing necroptosis. Comparison of MD simulations of the RIPK3 homo-amyloid and RIPK1/RIPK3 hetero-amyloid show that hetero-amyloid needs to have a different structure than the homo-amyloid (where a similar structure was found for the human hetero-amyloid), and so if RIPK3 is added in excess to the RIPK1/RIPK3 hetero-amyloid, we would expect to see a structural re-arrangement.

In summary:

Structural study

Mass-per-length measurements establish one protein molecule per cross- β unit

In-register parallel stacking verified based on 2D C-C spectra with [2-¹³C]-glycerol labeling (Ca-C β contacts within the same residue are primarily intermolecular, and several distinct intra-residue peaks are identified for 500 ms DARR. Mixed label 15N / ¹³C TEDOR confirms this finding).

10 unambiguous, non-sequential restraints plus torsional angle restraints, ambiguous restraints determine the experimental structure (in addition to force-field restraints, etc.).

Functional Studies

1st and 2nd β -strands required for Necroptosis (441VFNN444, 448VQIG451) but 3rd strand (455SLV457) shows only partial inhibition. These mutations inhibit the interaction between RIPK1 and RIPK3.

F442A, Q449A, L456A only decrease necroptosis slightly. (Q449 previously known as important to stabilizing fibril structure)

F442D, Q449D, L456D mutations inhibit necroptosis. Interaction between RIPK1/RIPK3 not changed. Correct fold of RIPK3 then affected. These mutations also slow down fibril growth in-vitro.

MD study

96 copies of each simulation for 50 ps. RIPK3 homo-amyloid and RIPK1/RIPK3 hetero-amyloid. Homo-amyloid stays in approximately the same configuration as SSNMR-determined structure (twist appears). Hetero-amyloid changes configuration, adopting a structure similar to the structure of human hetero-amyloid.

I found this to be an interesting article, elucidating the roles of two RHIM containing proteins, RIPK1 and RIPK3. I would suggest the revisions (mostly minor) suggested below.

AlbertA. Smith

Comments/Corrections

In line 265, the authors state "However, the change [single-site mutation] did ^{not} inhibit the cell necroptosis, probably by affecting the correct folding of RIPK3 fibril." But, the following paragraph

discusses that the mutations also change the fibril formation rate. How do the authors know that the correct fold is critical, as opposed to the fibril formation rate?

Of the various mutants, which form fibrils in solution?

There's some indication at line 77- "And these RIPK3 mutants [F442D, Q449D, L456D] still form the fibril in solution." Presumably, the Alanine mutants must also form fibrils. What about the 3 β -strand replacements with alanines? Perhaps it should be obvious to me, and all mutants fibrillize. But, if some of the mutants don't fibrillize, then that would of course be important information. Please clarify in paragraph starting line 241.

Line 78- "Combining these results, we could propose that both RIPK1-RIPK3 ^[11]_{SEP} binding and the RIPK3 amyloid formation are essential, but not sufficient for RIPK3 mediated necroptosis." and

Line 297- "The functional studies above indicate that the necroptosis pathway requires RIPK1-RIPK3 intermolecular interaction and RIPK1/RIPK3 hetero-oligomer formation."

I don't think we can really establish the requirement of RIPK1 and RIPK3 interaction from this study (a side note: line 78 might be over-interpreted to imply that the RIPK3 necroptosis pathway only occurs with RIPK1, although one finds in the literature that other RHIM containing proteins may be involved in the necroptosis pathway instead of RIPK1, e.g. review Dalton, doi: 10.1016/j.tcb.2015.01.001). The authors establish that mutation of regions in the RHIM domain can interfere with interaction between RIPK1 and RIPK3. But, such mutations also can influence fibrilization rates and fibril structure of RIPK3—a point the authors make, when noting that F442D, Q449D, and L456D mutations inhibited necroptosis but not interaction between RIPK1 and RIPK3. Then, I would rather refer to the literature when stating that RIPK1 is important for the necroptosis pathway, since I think the importance of RIPK1 has been established (but here, we would require some way of eliminating the RIPK1/RIPK3 interaction while still having wild-type RIPK3).

Given the proposal by the authors that one has initial formation of RIPK1/RIPK3 hetero-amyloid, followed by restructuring upon lengthening with RIPK3: why not try to simulate the restructuring with MD? The MD has provided a reasonable structure for the hetero-amyloid (agreeing fairly well with the SSNMR study by McDermott and co-workers). What happens if RIPK1 in this structure is then replaced by RIPK3? Can the transformation from the structure of the hetero-amyloid to the RIPK3 fibril be captured? This would seem to be a very relevant MD simulation to the proposed mechanism. Please include this simulation, or explain why it isn't possible. (it could also certainly be interesting to simulate varying ratios of RIPK1/RIPK3)

Are approximately equal amounts of RIPK1 and RIPK3 initially present, and then later an excess of RIPK3? Or is there always an RIPK3 excess? I find this to be a stumbling block when trying to picture the formation of RIPK3 in vivo.

For the MD simulations, can you establish that the structures are equilibrated after 50 ps? What do we see in the other 92 simulations? I'm not clear if the 96 repetitions are there because many of the simulations do not converge on the resulting structures, or if the transition between homo- and hetero-amyloid structures usually takes longer than 50 ps. It should be clear why there are 96 simulations, if we only see results of 4 of them.

Minor:

It is not clear from the main text whether the appearance of a twist in the MD simulations contradicts the experimental results (line 317). As I understand it, the structural model was not allowed to have a twist angle (line 622). This is a reasonable way to execute the structural calculation— there probably not enough restraints to calculate the twist angle unambiguously. Then, it means that appearance of the twist in simulation is not contradicted by the experiments, rather, it's information that the experiments couldn't provide. It might be worth pointing this out.

A table indicating the experimental type and samples used for each experiment used in structure determination would be useful (could also include experiments for assignment with samples). I think that intra-molecular contacts were established with a [¹³C,¹⁵N] labeled sample (DARR,

TEDOR), and the in-register parallel stacking is verified using a [2-13C]-glycerol labeling (DARR) and a mixed [13C,14N] and [12C, 15N] sample (TEDOR). But, a table would make this clearer.

Typos:

Line 80: 'which consists three ...' should be 'which consists of three ...'

Line 105: 'by dialysis the denatured protein' should be 'by dialysis of ...'

Line 260: 'a less' should be 'less' or 'a smaller'

Line 382-403: Why do we now use necrosis instead of necroptosis?

Line 642: 'pairs' not 'paris'

Reviewer #2 (Remarks to the Author):

The manuscript by Lu et al. reported amyloid structure of m-RIPK3 using solid-state NMR. MD simulations were also used to describe conformational changes from hetero RIPK1-RIPK3 amyloid to RIPK3 homo amyloid. In addition, they discussed functional implications of the amyloid structure and structural changes. The structural studies of homo and hetero amyloid of RIPK1 and RIPK3 may provide valuable information.

But the experimental results are too preliminary to be published. For example, the authors claimed that "the structural integrity of RIPK3 fibril with three β -strands is necessary for the signaling" in Abstract (26 – 27). They drew the conclusion based on the different aggregation kinetics and ThT fluorescence intensity of WT and mutant RIPK3 fibrils. However, the different aggregation kinetics probed by ThT fluorescence and different ThT fluorescence intensity of the fibrils do NOT guarantee at all that the mutant fibrils have different structures. The authors should use solid-state NMR to confirm that the mutant fibrils have distinct molecular conformations from WT fibrils.

Secondly, the authors used MD simulations (50 ps) to probe structural changes of hetero RIPK1-PIP3 amyloid. However, they built the hetero amyloid model by simply replacing the RIPK3 sequence by the RIPK1 in the RIPK3 amyloid structure. But this arbitrary structure does not represent a real hetero-amyloid structure and the conformational changes they observed are not meaningful. Fifty ps timescale is also too short to probe the conformational changes.

Other issues:

1. Authors claimed that they solved RIPK3 amyloid structure. However, a C-terminal fragment containing the RHIM domain (409-486) was used for the structural studies. It is not clear whether the 486-residue full-length protein would form the same amyloid core consisting of only about 20 residues that they observed in the RHIM domain fibrils. Authors could have tried to induce fibril formation of the full-length protein using the RHIM-domain fibrils (seeding) to make sure they have same amyloid core.

2. What are the linewidths of 13C and 15N NMR of the fibrils. Are there any additional NMR resonances in the 2D and 3D spectra?

3. Authors should use a mixture of the samples (13C/15N-labeled and unlabeled) to make sure that the cross-peaks they observed (described in 164 – 166 in the manuscript) are from intramolecular interactions.

4. How many ThT fluorescence experiments were carried out in Figure S7 (errors should be indicated). Experimental conditions for fibril formation (protein concentration, incubation times etc.) should be described in Figure S7.

Reviewer #3 (Remarks to the Author):

The manuscript of Wu, Hu et al. seeks to bridge a major gap in understanding relating to how RIP3 can assemble into high molecular weight platforms for necroptosis signaling via the short motif termed the RHIM that is known to form amyloid fibrils. One of the mysteries arising from the Mompean et al. Cell 2018 paper was if RIP1 and RIP3 prefer an alternating arrangement, how does RIP3 self-assemble and auto-activate? The structure reported in the present submission provides some answers for this conundrum. The ssNMR structure and accompanying mutagenesis and MD indicate that, in mouse RIP3, the RHIM motif adopts an alternative conformation and relies on flanking sequences to adopt an "N" conformation. This connects the dots between prior reports (some of which should be referenced by the authors), but would benefit from validation of two key facets: (1) is the observation more broadly applicable to other species' RIP3 e.g. in human cells and (2) stronger support for mutations flanking the core RHIM tetrad impacting necroptosis in cellular studies.

Major points

1. The points of difference with earlier reported RHIM structures are not well communicated. The comparative structure figures are shown in supplement, but should be in the main text. Likewise the authors should present a sequence alignment in the main text and comment on the sequence conservation of the key residues. If they are non-homologous, further MD is warranted to examine whether a comparable structure might be observed in human RIP3. In the abstract, it would be beneficial for the authors to describe how the presented structure differs from the Mompean structure rather than simply saying they differ.

2. Relatedly, whether the observed structure would be present in other species, such as humans, is not clear from the manuscript. It would be of less interest if the structure were confined to mouse proteins and it is the species difference that is the basis for the differences with the Mompean structure. This is a crucial point that the authors need to address.

3. In relation to the differences in species: differences between the necroptosis pathway have been observed in other studies, especially between mouse and human signaling e.g. Petrie Nature Commun 2018, Tanzer CDD 2016, Davies CDD 2018, Davies Nature Commun 2020. It is important that the authors consider these differences in the context of their findings. These studies highlight the importance of establishing whether the RIP3 conformation reported is specific to mice, or whether it is illustrative of a widely conserved mechanism. Much of the introductory literature pertains to human necroptosis signaling (refs. 5, 7, 8) with incomplete coverage of the relevant mouse literature (e.g. ref. 6 Rodriguez CDD is pre-dated by Murphy Immunity 2013 and Tanzer Biochem J 2015), which seems like an oversight considering this manuscript is solely focused on the mouse system.

4. The coverage of relevant literature in the introduction is very thin and some of the ideas presented are either wrong or outmoded. This is surprising considering the seminal work of one of the corresponding authors in the necroptosis field. For example, the authors propose the RIP1:RIP3 platform is called a signalosome. This is not the convention; it is typically called a necrosome. The authors describe the assembly of MLKL into higher order complexes via coiled coils. This is an outmoded idea. There are structural studies of MLKL that provide a clearer picture of how these assemble. The coverage of RHIM knowledge is far from complete also – e.g. Pham EMBO Rep 2018 is overlooked. Line 39- RHIM longhand is incorrect. I feel it is important that the authors familiarize themselves with the current state of the field and present a more contemporary view of the pathway. I realise it is a fast moving area but there are numerous authoritative reviews that should have led the authors to a more contemporary take on the pathway.

5. Cellular studies. The authors use an overexpression system to examine the effects of RIP3 mutants in NIH-3T3 cells in addition to some coexpression studies of RIP1 and RIP3 in 293T. I have three major concerns with these analyses:

a. NIH3T3 cells are necroptotic when treated with TSZ (e.g. Müller et al., CMLS 2017 and the cited Sun et al. Cell 2012 on which the co-corresponding Huayi Wang is an author), which is contrary to what is stated in the manuscript. The control experiment in which untransduced NIH3T3 cells are treated with TSZ would have revealed this. I am unclear why He Cell 2009 was cited in reference to NIH3T3 cells. I would caution against this approach because examining mutant RIP3 when

endogenous RIP3 is present could lead to artefacts. Examining reconstitution of necroptosis by wild-type RIP3 and the RHIM mutants in a relevant cell type like RIP3 KO mouse fibroblasts would be more compelling.

b. 293T was used to test RIP1:RIP3 interaction but this is not a necroptotic cell line. It would be more robust to express RIP3 mutants in RIP3 KO mouse fibroblasts or L929 cells, which would be consistent with some of the more reliable studies in the literature. Such an approach would be the ideal platform to examine interaction of the tagged RIP3 wild-type or extended RHIM mutant with the endogenous RIP1.

c. The authors do not examine the effect of RIP3 mutation on interaction with RIP1 in cells. While not a necroptotic line, 293T would lend themselves to such a study because it would obviate the role of RIP1 in nucleating higher order assemblies and thus RIP1 would not contribute to any RIP3 higher order species. This would require coexpression of wild-type RIP3 and the mutants different tags for IP studies, which would be very feasible.

6. Relatedly, on line 245 the authors state that RIPK1 and MLKL are ubiquitously expressed in commonly used cell lines. This is a misrepresentation of the reality. Those used for necroptosis are used because of their ubiquitous RIP1, RIP3 and MLKL expression; not all cell lines are appropriate for these studies. 293T for example are not appropriate and arguably the most commonly used line in all of the literature. I suggest the authors rephrase this section to more accurately capture which cell types are suitable for necroptosis studies.

7. Reproducibility. The authors do not state how many repeats each experiment represents in their legends. The authors present viability data as mean and "SD of duplicate wells". Duplicates are not sufficient to enable statistical analyses and I would expect errors to represent the variation between multiple experiments.

8. In the structure figures, the authors should label the features described in the main text, such as beta-arch, in addition to the strands of interest.

9. Line 297. The authors do not validate that RIP1:RIP3 interaction are required for necroptosis, unlike what is claimed.

10. The role of the kinase domains in RIP1 and RIP3 in dictating the higher order assemblies and how these might contribute to regulation of necrosome/RHIM mediated oligomers is not considered but should be an integral part of any model

11. Line 382 and related figures. Because the coIP interaction data are qualitative, the authors have not ruled out a kinetic defect in RIP1 interaction with the flanking RHIM RIP3 mutants, so this requires some caution in interpretation. Alternatively, the authors should exclude a defect in interaction/kinetic assembly by performing time course experiments.

Minor points

The manuscript would benefit from critical proofreading and/or English editing. There are many errors that distracted me during reading the work.

I find it strange that the manuscript is submitted formatted for another publisher. Regardless, the STAR Methods table is incomplete because the catalog numbers of antibodies are an important consideration when many in the field are not fit for purpose.

Figure 1E label. White text would allow the E label to be more readily seen.

Line 135. What was protein labelled with?

Line 183. Name valines.

Figure 7 legend. Software for MD needs to be described.

REVIEWER COMMENTS

Reviewer #1 (Remarks to the Author):

Lu and co-workers investigate the role of mouse RIPK3 in cell necroptosis. This study includes a functional investigation of the roles of RIPK1 and RIPK3, a solid-state NMR based structural study of mouse RIPK3 fibrils, and an MD study to establish the stability of RIPK3 fibrils, but the instability of the RIPK1/RIPK3 hetero-amyloid when forced to take the same structure as the RIPK3 fibrils. The hetero-amyloid adopts a structure similar to a previously determined structure of human RIPK1/RIPK3 hetero-amyloids.

The RIPK3 structure, established based on solid-state NMR and MPL measurements, is well-supported. Functional studies establish the relative importance of the β -strands found in RIPK3 in inducing necroptosis. Comparison of MD simulations of the RIPK3 homo-amyloid and RIPK1/RIPK3 hetero-amyloid show that hetero-amyloid needs to have a different structure than the homo-amyloid (where a similar structure was found for the human hetero-amyloid), and so if RIPK3 is added in excess to the RIPK1/RIPK3 hetero-amyloid, we would expect to see a structural re-arrangement.

In summary:

Structural study

Mass-per-length measurements establish one protein molecule per cross- β unit

In-register parallel stacking verified based on 2D C-C spectra with [2- ^{13}C]-glycerol labeling ($\text{C}\alpha$ - $\text{C}\beta$ contacts within the same residue are primarily intermolecular, and several distinct intra-residue peaks are identified for 500 ms DARR. Mixed label ^{15}N / ^{13}C TEDOR confirms this finding).

10 unambiguous, non-sequential restraints plus torsional angle restraints, ambiguous restraints determine the experimental structure (in addition to force-field restraints, etc.).

Functional Studies

1st and 2nd β -strands required for Necroptosis (441VFNN444, 448VQIG451) but 3rd strand (455SLV457) shows only partial inhibition. These mutations inhibit the interaction between RIPK1 and RIPK3.

F442A, Q449A, L456A only decrease necroptosis slightly. (Q449 previously known as important to stabilizing fibril structure)

F442D, Q449D, L456D mutations inhibit necroptosis. Interaction between RIPK1/RIPK3 not changed. Correct fold of RIPK3 then affected. These mutations also slow down fibril growth in-vitro.

MD study

96 copies of each simulation for 50 ps. RIPK3 homo-amyloid and RIPK1/RIPK3 hetero-amyloid. Homo-amyloid stays in approximately the same configuration as SSNMR-determined structure (twist appears). Hetero-amyloid changes configuration, adopting a structure similar to the structure of human hetero-amyloid.

I found this to be an interesting article, elucidating the roles of two RHIM containing proteins, RIPK1 and RIPK3. I would suggest the revisions (mostly minor) suggested below.

Comments/Corrections

In line 265, the authors state "However, the change [single-site mutation] did inhibit the cell necroptosis, probably by affecting the correct folding of RIPK3 fibril." But, the following paragraph discusses that the mutations also change the fibril formation rate. How do the authors know that the correct fold is critical, as opposed to the fibril formation rate?

Of the various mutants, which form fibrils in solution?

There's some indication at line 77- "And these RIPK3 mutants [F442D, Q449D, L456D] still form the fibril in solution." Presumably, the Alanine mutants must also form fibrils. What about the 3 β -strand replacements with alanines? Perhaps it should be obvious to me, and all mutants fibrilize. But, if some of the mutants don't fibrilize, then that would of course be important information. Please clarify in paragraph starting line 241.

Line 78- "Combining these results, we could propose that both RIPK1-RIPK3 binding and the RIPK3 amyloid formation are essential, but not sufficient for RIPK3 mediated necroptosis."

and

Line 297- "The functional studies above indicate that the necroptosis pathway requires RIPK1-RIPK3 intermolecular interaction and RIPK1/RIPK3 hetero-oligomer formation."

I don't think we can really establish the requirement of RIPK1 and RIPK3 interaction from this study (a side note: line 78 might be over-interpreted to imply that the RIPK3 necroptosis pathway only occurs with RIPK1, although one finds in the literature that other RHIM containing proteins may be involved in the necroptosis pathway instead of RIPK1, e.g. review Dalton, doi: 10.1016/j.tcb.2015.01.001). The authors establish that mutation of regions in the RHIM domain can interfere with interaction between RIPK1 and RIPK3. But, such mutations also can influence fibrilization rates and fibril structure of RIPK3—a point the authors make, when noting that F442D, Q449D, and L456D mutations inhibited necroptosis but not interaction between RIPK1 and RIPK3. Then, I would rather refer to the literature when stating that RIPK1 is important for the necroptosis pathway, since I think the importance of RIPK1 has been established (but here, we would require some way of eliminating the RIPK1/RIPK3 interaction while still having wild-type RIPK3).

Given the proposal by the authors that one has initial formation of RIPK1/RIPK3 hetero-amyloid, followed by restructuring upon lengthening with RIPK3: why not try to simulate the restructuring with MD? The MD has provided a reasonable structure for the hetero-amyloid (agreeing fairly well with the SSNMR study by McDermott and co-workers). What happens if RIPK1 in this structure is then replaced by RIPK3? Can the transformation from the structure of the hetero-amyloid to the RIPK3 fibril be captured? This would seem to be a very relevant MD simulation to the proposed mechanism. Please include this simulation, or explain why it isn't possible. (it could also certainly be interesting to simulate varying ratios of RIPK1/RIPK3)

Are approximately equal amounts of RIPK1 and RIPK3 initially present, and then later an excess of RIPK3? Or is there always an RIPK3 excess? I find this to be a stumbling block when trying to picture the formation of RIPK3 in vivo.

For the MD simulations, can you establish that the structures are equilibrated after 50 ps? What do we see in the other 92 simulations? I'm not clear if the 96 repetitions are there because many of the simulations do not converge on the resulting structures, or if the transition between homo- and hetero-amyloid structures usually takes longer than 50 ps. It should be clear why there are 96 simulations, if we only see results of 4 of them.

Minor:

It is not clear from the main text whether the appearance of a twist in the MD simulations contradicts the experimental results (line 317). As I understand it, the structural model was not allowed to have a twist angle (line 622). This is a reasonable way to execute the structural calculation— there probably not enough restraints to calculate the twist angle unambiguously. Then, it means that appearance of the twist in simulation is not contradicted by the experiments, rather, it's information that the experiments couldn't provide. It might be worth pointing this out.

A table indicating the experimental type and samples used for each experiment used in structure determination would be useful (could also include experiments for assignment with samples). I think that intra-molecular contacts were established with a [13C,15N] labeled sample (DARR, TEDOR), and the in-register parallel stacking is verified using a [2-13C]-glycerol labeling (DARR) and a mixed [13C,14N] and [12C, 15N] sample (TEDOR). But, a table would make this clearer.

Typos:

Line 80: 'which consists three ...' should be 'which consists of three ...'

Line 105: 'by dialysis the denatured protein' should be 'by dialysis of ...'

Line 260: 'a less' should be 'less' or 'a smaller'

Line 382-403: Why do we now use necrosis instead of necroptosis?

Line 642: 'pairs' not 'paris'

Response to Reviewer #1

Thanks for your detailed and exact summary for our work and great interests to the article.

Comment 1:

“In line 265, the authors state “However, the change [single-site mutation] did inhibit the cell necroptosis, probably by affecting the correct folding of RIPK3 fibril.” But, the following paragraph discusses that the mutations also change the fibril formation rate. How do the authors know that the correct fold is critical, as opposed to the fibril formation rate?”

Response:

We agree with the reviewer in this point. The *in vitro* studies on the mutants (F442D, Q449D and L456D) were carried out on very simple solution conditions with pure proteins. It served as basic characterizations of the mutants. It showed that the fibril formation rates as well as the fibril conformations were affected by the mutations. The cell environment is more complex, involving many interacting components. The cell necroptosis assay is also a much longer process compared to the *in vitro* fibril growth assay. However, only with the correct RIPK3 fibril assembly formed in a timely manner, will the cell necroptosis happen. The observation on the mutation inhibited the cell necroptosis could be explained by fibril formation rates or the conformation changes. Therefore, we think it is better to delete the sentence in line 265 “probably by affecting the correct folding of RIPK3 fibril.” to avoid the confusion.

Comment 2:

“Of the various mutants, which form fibrils in solution?”

There’s some indication at line 77- “And these RIPK3 mutants [F442D, Q449D, L456D] still form the fibril in solution.” Presumably, the Alanine mutants must also form fibrils. What about the 3 β -strand replacements with alanines? Perhaps it should be obvious to me, and all mutants fibrilize. But, if some of the mutants don’t fibrilize, then that would of course be important information. Please clarify in paragraph starting line 241.”

Response:

From the ThT binding assay, EM images and x-ray images, we confirmed that RIPK3 mutants [F442D, Q449D, L456D] still form fibrils in solution. Assuming a less disruptive change to Alanine compared to Aspartic acid, we think the mutants [F442A, Q449A, L456A] would form fibrils. We carried out additional experiments and expressed the single-site RIPK3 alanine mutant [L456A]. We discovered that L456A indeed aggregated into fibrils. Additional test on F442A and Q449A were not carried out.

One TEM image and the ThT fluorescence intensity curve of L456A mouse RIPK3 fibril.

As for the mutants with β -strand replacements with alanines, we were not able to induce the expression of these mutants using the same plasmid construction as the wild type proteins in *E. Coli*. Therefore, the fibril formation was not tested. But the article by Chi LL Pham (Viral M45 and necroptosis-associated proteins form heteromeric amyloid assemblies. 2019 EMBO Rep 20 (2). DOI: 10.15252/embr.201846518.) has shown the ThT fluorescence curve and the EM images of assemblies formed by His-Ub-RIPK3₃₈₇₋₅₁₈AAAA (human RIPK3), of which the center RHIM sequence were replaced by four alanines. They further demonstrated that “the mutation of the VQVG RHIM tetrad of RIPK3 to AAAA does not abolish the ability of this protein to self-assemble and the mutant RIPK3 construct retains the ability to interact with the M45 RHIM”. Therefore, we would think the mutants with the individual β -strand replacement could still form fibrils. Considering the immunoprecipitation results in our work showed the abolish of the interaction between RIPK1 and RIPK3 when RIPK3 1st or 2nd β -strand was changed to Alanines, the disruption of cell necroptosis is at this step before RIPK3 self-assembled fibril is formed.

Comment 3:

“Line 78- “Combining these results, we could propose that both RIPK1-RIPK3 binding and the RIPK3 amyloid formation are essential, but not sufficient for RIPK3 mediated necroptosis.” And Line 297- “The functional studies above indicate that the necroptosis pathway requires RIPK1-RIPK3 intermolecular interaction and RIPK1/RIPK3 hetero-oligomer formation.” I don’t think we can really establish the requirement of RIPK1 and RIPK3 interaction from this study (a side note: line 78 might be over-interpreted to imply that the RIPK3 necroptosis pathway only occurs with RIPK1, although one finds in the literature that other RHIM containing proteins may be involved in the necroptosis pathway instead of RIPK1, e.g. review Dalton, doi: 10.1016/j.tcb.2015.01.001). The authors establish that mutation of regions in the RHIM domain can interfere with interaction between RIPK1 and RIPK3. But, such mutations also can influence fibrilization rates and fibril structure of RIPK3—a point the authors make, when noting that F442D, Q449D, and L456D mutations inhibited necroptosis but not interaction between RIPK1 and RIPK3. Then, I would rather refer to the literature when stating that RIPK1 is important for the necroptosis pathway, since I think the importance of RIPK1 has been established (but here, we would require some way of eliminating the RIPK1/RIPK3 interaction while still having wild-type RIPK3).”

Response:

Thanks for the comment. In our work, we just focused on the classic necroptosis pathway which has been widely recognized that RIPK1 can bind with RIPK3 through RHIM domain to form a necrosome when activated by the TNF α factors. In our experiment, z-VAD and Smac were used as the mimetic compounds to induce this classic necroptosis pathway which is an effective way as reported previously. We agree that the sentences in Line 78 and 279 were not appropriate and therefore changed to emphasize the TNF α -induced necroptosis. References were also added at line 279.

The reviewer is correct that other RHIM domain containing proteins like DAI and TRIF also participate in the necroptosis process by activating RIPK3. But TNF we used in this study could not trigger DAI/TRIF-dependent necroptosis. The figure below shows an overview of different pathways leading to necroptosis in the cell (cited from Trends in cell biology, october 2016, Vol.26, No.10.)

Comment 4:

“Given the proposal by the authors that one has initial formation of RIPK1/RIPK3 hetero-amyloid, followed by restructuring upon lengthening with RIPK3: why not try to simulate the restructuring with MD? The MD has provided a reasonable structure for the hetero-amyloid (agreeing fairly well with the SSNMR study by McDermott and co-workers). What happens if RIPK1 in this structure is then replaced by RIPK3? Can the transformation from the structure of the hetero-amyloid to the RIPK3 fibril be captured? This would seem to be a very relevant MD simulation to the proposed mechanism. Please include this simulation, or explain why it isn’t possible. (It could also certainly be interesting to simulate varying ratios of RIPK1/RIPK3).”

Response:

It is reasonable to think of a MD study using the hetero-amyloid by McDermott as a starting structure and replacing RIPK1 by RIPK3. We indeed tried at the beginning but the structure only showed a little loosening during the time we observed. It did not show a big structural transformation to clearly demonstrate a similarity to our RIPK3 structure. We have the following difficulty to improve the MD study. The published RIPK1/3 structure is for human and the sequence length in core structure is also shorter than our RIPK3 fibrils. So when we replace RIPK1 by RIPK3, the only good replacement is around RHIM core region. The 1st β -strand is missing and 3rd β -strand is shorter, unstructured and not aligned. Therefore, it would take a much longer time for these regions to adopt ordered β -strand conformation with a correct β -sheet registration. Besides, for this structural transformation to happen, a correct starting structure is very important. However, for those missing residues, the starting conformation is random. Therefore, we could only focus on our own RIPK3 fibril structure, using it as a start structure for MD studies.

Comment 5:

“Are approximately equal amounts of RIPK1 and RIPK3 initially present, and then later an excess of RIPK3? Or is there always an RIPK3 excess? I find this to be a stumbling block when trying to picture the formation of RIPK3 in vivo.”

Response:

It's actually unknown so far, the amounts of RIPK1 and RIPK3 initially present in cells. In normal physiology condition, RIPK3 is auto-inhibited and freely scattered in cell, usually no RIPK3 fibril formed. There won't be the RIPK1/3 amyloid fibril formation without the activation of TNF α factors as proved by Jixi Li in 2012 (The RIP1/RIP3 necrosome forms a functional amyloid signaling complex required for programmed necrosis. *Cell* 150 (2), pp. 339–350. DOI: 10.1016/j.cell.2012.06.019.) Furthermore, in 2014, Jiahuai Han' lab (Distinct roles of RIP1–RIP3 hetero- and RIP3–RIP3 homo-interaction in mediating necroptosis. *Cell Death Differ* 21 (11), pp. 1709–1720. DOI: 10.1038/cdd.2014.77) reported that “RIPK1-RIPK3 heterodimer itself cannot induce necroptosis, the RIPK1-

RIPK3 heterodimeric amyloid fibril is unlikely to directly propagate necroptosis". They proposed that the event after RIPK1-RIPK3 heterodimeric amyloid fibril is the recruitment of RIPK3. The RIPK3 also has to be activated during the fibril formation. So, we think in the cells, it is not the relative amount of RIPK1 or RIPK3 that matters, it is the correct regulation and activation of the proteins that matters. The TNF α factors only can induce RIPK1 mediated necroptosis pathway, and promote RIPK1 self-assembly formation first even RIPK3 is also present in the cells. RIPK1/RIPK3 heterodimeric amyloid fibril could be thought of as fibril seeds to initiate the RIPK3 fibrils. The phrase "more RIPK3 coming" indicated in Figure 7 (current figure 8) may be confusing. To avoid this confusion, we have replaced these words by "recruiting RIPK3".

Comment 6:

"For the MD simulations, can you establish that the structures are equilibrated after 50 ps? What do we see in the other 92 simulations? I'm not clear if the 96 repetitions are there because many of the simulations do not converge on the resulting structures, or if the transition between homo- and hetero-amyloid structures usually takes longer than 50 ps. It should be clear why there are 96 simulations, if we only see results of 4 of them."

Response:

In this paper, the MD study was carried out using xplor-nih. It could be considered as an additional equilibrium run after the structural calculation for RIPK3 fibrils. The 96 repetitions were run for the statistic purpose, all reached to a converge structure and displayed the equilibrium energies after 50ps. The potential energy changes for the three different MD studies is shown in figure below, also in supplement figure S10.

Figure S10 (A) The energy including all potential energy terms (kcal/mol) changes along the 50 ps MD runs. The MD study of mouse RIPK3 fibril is in green; The MD runs for mouse RIPK1/RIPK3 fibril are shown in black and red for two different RIPK1 and RIPK3 alignments. The alignment 1 (black) corresponds to the alignment shown in figure 8C while the alignment 2 (red) corresponds to the alignment in figure S9. All three are from the structures with the minimum energy among the 96 repeats.

There are structural differences among the 96 repetitions. The figure below showed the comparison between the best one and worst one for RIPK3 fibril MD study and RIPK1/RIPK3 fibrils. They are also added as supplement figure S10. From the figure, we could see the difference is not significant.

Figure S10 (B) The structural comparison between the best structure (solid color) and the worse structure (translucent) from the 96 repetitions of MD studies.

Comment 7:

“It is not clear from the main text whether the appearance of a twist in the MD simulations contradicts the experimental results (line 317). As I understand it, the structural model was not allowed to have a twist angle (line 622). This is a reasonable way to execute the structural calculation– there probably not enough restraints to calculate the twist angle unambiguously. Then, it means that appearance of the twist

in simulation is not contradicted by the experiments, rather, it's an information that the experiments couldn't provide. It might be worth pointing this out."

Response:

We appreciate the reviewer's comment on the twist angle. Indeed, the twist angle is consistent with our experimental results. We calculate the structure without using the twist angle information. The MD run finally gave the fibril a twist angle about $6.5^{\circ} \pm 1^{\circ}$. We later managed to take better fibril images using AFM. We can obtain the twist angle from the AFM images by measuring the periodic height changes along the fibril growth direction. Through measurement, we got a pitch value of 28.5 nm, which indicates the twist angle $= 360 / (28.5 / 0.48) = 6.1^{\circ}$. We found that the relaxed mouse RIPK3 structures have similar twist angle with the data we got from the AFM images. The information was added to the paper.

Figure S11 (A) One AFM image of mouse RIPK3. (B) The expanded view of the box highlighted in A. (C) The periodic height changes along the line shown in B.

Comment 8:

"A table indicating the experimental type and samples used for each experiment used in structure determination would be useful (could also include experiments for

assignment with samples). I think that intra-molecular contacts were established with a [13C, 15N] labeled sample (DARR, TEDOR), and the in-register parallel stacking is verified using a [2-13C]-glycerol labeling (DARR) and a mixed [13C, 14N] and [12C, 15N] sample (TEDOR). But, a table would make this clearer.”

Response:

A table with all the NMR experiments is added to the supplement information and shown below.

Table S1. Experimental parameters in SSNMR for mRIPK3 fibril structure determination.

sample	MAS (KHz)	Experiment	Recycle delay (s)	tppm (kHz)	Mixing time (ms)	Cross-polarization time (ms)	Number of scans	Total Time (h)	Temperature (K)
	15	DARR	2	83.3	50	1.5	36	10.8	303
	15	DARR	2	83.3	200	1.5	36	11.5	303
	15	DARR	2	83.3	500	1.5	72	29.0	303
	15	2d-NCaCX	2	83.3	50	1.5, 5.5	256	7.2	303
	15	2d-NCocX	2	83.3	50	1, 4.5	256	7.2	303
mRIPK3 (uniformly ¹³ C, ¹⁵ N labeled)	15	3d-NCACX	2	83.3	50	1.5, 5.5	64	96	303
	15	3d-NCOCX	2	83.3	50	1.5, 5.5	64	59.5	303
	15	TEDOR	2	83.3	6.4	1.5	256	18.5	303
	15	TEDOR	2	83.3	8.5	1.5	256	18.5	303
mRIPK3 (2- ¹³ C glycecol)	15	DARR	2	100	50	1.5	48	14.3	303
	15	DARR	2	100	200	1.5	80	25.6	303
and	15	DARR	2	100	500	1.5	96	34.8	303
mRIPK3 (1,3- ¹³ C glycecol)	15	2d-NCaCX	2	100	50	1.5, 5.5	256	19.0	303
	15	2d-NCocX	2	100	50	1.5, 4.5	256	19.0	303
	15	TEDOR	2	100	3.2, 6.4, 8.5	1.5	256	18.5	303
mRIPK3 (1:1 mixed ¹³ C/ ¹⁵ N labeled)	15	DARR	2	100	50	1.5	64	19.0	303
	15	DARR	2	100	200	1.5	64	20.5	303
	15	DARR	2	100	500	1.5	104	37.8	303
	15	TEDOR	2	83.3	6.4,8.5	1.5	256	18.5	252
	15	NhhC	2	83.3			512	23.3	252

Comment 9:**“Typos:*****Line 80: ‘which consists three ...’ should be ‘which consists of three ...’***

We have corrected in line 80 ‘which consists three ...’ to ‘which consists of three ...’

Line 105: ‘by dialysis the denatured protein’ should be ‘by dialysis of ...’

We have corrected in line 105 ‘by dialysis the denatured protein’ to ‘by dialysis of ...’

Line 260: ‘a less’ should be ‘less’ or ‘a smaller’

We have corrected in line 260 ‘a less’ to ‘a smaller’.

Line 382-403: Why do we now use necrosis instead of necroptosis?

We’ve replaced it with ‘necroptosis’.

Line 642: ‘pairs’ not ‘paris’”

We’ve replaced ‘paris’ with ‘pairs’.

Reviewer #2 (Remarks to the Author):

The manuscript by Lu et al. reported amyloid structure of m-RIPK3 using solid-state NMR. MD simulations were also used to describe conformational changes from hetero RIPK1-RIPK3 amyloid to RIPK3 homo amyloid. In addition, they discussed functional implications of the amyloid structure and structural changes. The structural studies of homo and hetero amyloid of RIPK1 and RIPK3 may provide valuable information.

But the experimental results are too preliminary to be published. For example, the authors claimed that “the structural integrity of RIPK3 fibril with three b-strands is necessary for the signaling” in Abstract (26 – 27). They drew the conclusion based on the different aggregation kinetics and ThT fluorescence intensity of WT and mutant RIPK3 fibrils. However, the different aggregation kinetics probed by ThT fluorescence and different ThT fluorescence intensity of the fibrils do NOT guarantee at all that the mutant fibrils have different structures. The authors should use solid-state NMR to confirm that the mutant fibrils have distinct molecular conformations from WT fibrils.

Secondly, the authors used MD simulations (50 ps) to probe structural changes of hetero RIPK1-RIPK3 amyloid. However, they built the hetero amyloid model by simply replacing the RIPK3 sequence by the RIPK1 in the RIPK3 amyloid structure. But this arbitrary structure does not represent a real hetero-amyloid structure and the conformational changes they observed are not meaningful. Fifty ps timescale is also too short to probe the conformational changes.

Other issues:

1. Authors claimed that they solved RIPK3 amyloid structure. However, a C-terminal fragment containing the RHIM domain (409-486) was used for the structural studies. It is not clear whether the 486-residue full-length protein would form the same amyloid core consisting of only about 20 residues that they observed in the RHIM domain fibrils. Authors could have tried to induce fibril formation of the full-length protein using the RHIM-domain fibrils (seeding) to make sure they have same amyloid core.
2. What are the linewidths of ¹³C and ¹⁵N NMR of the fibrils. Are there any additional NMR resonances in the 2D and 3D spectra?
3. Authors should use a mixture of the samples (¹³C/¹⁵N-labeled and unlabeled) to make sure that the cross-peaks they observed (described in 164 – 166 in the manuscript) are from intramolecular interactions.
4. How many ThT fluorescence experiments were carried out in Figure S7 (errors should be indicated). Experimental conditions for fibril formation (protein concentration, incubation times etc.) should be described in Figure S7.

Answers to Reviewer #2

Comment 1:

“The experimental results are too preliminary to be published. For example, the authors claimed that “the structural integrity of RIPK3 fibril with three b-strands is necessary for the signaling” in Abstract (26 – 27). They drew the conclusion based on the different aggregation kinetics and ThT fluorescence intensity of WT and mutant RIPK3 fibrils. However, the different aggregation kinetics probed by ThT fluorescence and different ThT fluorescence intensity of the fibrils do NOT guarantee at all that the mutant fibrils have different structures. The authors should use solid-state NMR to confirm that the mutant fibrils have distinct molecular conformations from WT fibrils.”

Response:

Thanks for your comment. The ThT fluorescence changes and kinetics strongly suggest the mutant fibrils would be different from the wild type. In addition, we have performed the SSNMR experiment for mutant fibrils [Q449D, L456D] (figure shown below). F442D has a lower protein yield, not enough for isotopic labeling and SSNMR studies. We find the F442C α -C β , S455C β -C α , TC β -C α , cross-peaks are among the peaks that still remain in the mutant spectra. The peaks for C445, S446 and E447 are disappeared for both mutants suggesting the disordering for the loop connecting 1st and 2nd β -strand. I450 C α -C β shifts to a similar position for both mutants, indicating a structural change at the RHIM core region. The peaks for N and V residues shift with lower resolutions than before. Our results confirmed that the single-site mutation on RIPK3 (Q449D and L456D) cause changes in the fibril structure. The information was added to the paper and supplement figure S8.

Figure S8 Comparison of 2D ^{13}C - ^{13}C correlation spectra of uniformly ^{13}C -labeled mouse RIPK3 fibrils and its mutants. Wild type (red), mutant Q449D (green), L456D (blue). The assignment in the spectra is for the wild type.

Comment 2:

“Secondly, the authors used MD simulations (50 ps) to probe structural changes of hetero RIPK1-PIP3 amyloid. However, they built the hetero amyloid model by simply replacing the RIPK3 sequence by the RIPK1 in the RIPK3 amyloid structure. But this arbitrary structure does not represent a real hetero-amyloid structure and the conformational changes they observed are not meaningful. Fifty ps timescale is also too short to probe the conformational changes.”

Response:

Thanks for the comment. We used XPLOR-NIH to carry out the MD studies. We obtained the energy equilibrium states within the 50 ps for all three different MD studies (The figure is shown below, also added in the supplement information figure S10 in the paper). RIPK3 fibril relaxation study by MD provided a meaningful structure. The MD run finally gave the fibril a twist angle about $6.5^\circ \pm 1^\circ$. We later managed to take better fibril images using AFM. We can obtain the twist angle from the AFM images by measuring the periodic height

changes along the fibril growth direction. Through measurement, we got a pitch value of 28.5 nm, which indicates the twist angle= $360/(28.5/0.48)=6.1^\circ$. We found that the relaxed mouse RIPK3 structures have similar twist angle with the data we got from the AFM images. The information was added to the paper (supplement figure S11). We agree that RIPK1/RIPK3 hetero-amyloid structure is an arbitrary structure based on modeling assuming that the published human hetero-amyloid NMR structure would be similar to mouse hetero-amyloid fibril. However, the purpose of this studies is to demonstrate the dynamic difference between RIPK3 fibril and the hypothetic RIPK1/RIPK3 fibrils, which clearly showed that RIPK3 fibril with “N” shape is stable.

Figure S10 (A) The energy including all potential energy terms (kcal/mol) changes along the 50 ps MD runs. The MD study of mouse RIPK3 fibril is in green; The MD runs for mouse RIPK1/RIPK3 fibril are shown in black and red for two different RIPK1 and RIPK3 alignments. The alignment 1 (black) corresponds to the alignment shown in figure 8C while the alignment 2 (red) corresponds to the alignment in figure S9. All three are from the structures with the minimum energy among the 96 repeats.

Figure S11 (A) AFM images of mouse RIPK3. (B) The expanded view of the box highlighted in A. (C) The periodic height changes along the line shown in B.

Comment 3:

“Authors claimed that they solved RIPK3 amyloid structure. However, a C-terminal fragment containing the RHIM domain (409-486) was used for the structural studies. It is not clear whether the 486-residue full-length protein would form the same amyloid core consisting of only about 20 residues that they observed in the RHIM domain fibrils. Authors could have tried to induce fibril formation of the full-length protein using the RHIM-domain fibrils (seeding) to make sure they have same amyloid core.”

Response:

We agree with you that it is good to know whether the 486-residue full-length protein form the same amyloid structure. In 2012, Hao Wu’s lab has studied the RIPK1/RIPK3 complex using the full-length proteins. Upon limited proteolysis to remove the flanking domains by subtilisin, clear fibril TEM images could be seen, similar to RIPK1/RIPK3 formed by the

RHIM domain (The RIP1/RIP3 necrosome forms a functional amyloid signaling complex required for programmed necrosis. *Cell* 150 (2), pp. 339–350. DOI: 10.1016/j.cell.2012.06.019). However, the SSNMR studies on human RIPK1/RIPK3 hetero-amyloid were still carried out using the C-terminal domain in their work. The full-length protein was expressed in insect cells. So far it is still difficult to obtain enough amount of the purified protein to carry out the NMR structural studies. We hypothesize that the full-length protein would adopt a similar fibril structure as the C-terminal domain of RIPK3. We think adding seeds to the full-length is a good idea. We will further study this issue.

Comment 4:

“What are the linewidths of ¹³C and ¹⁵N NMR of the fibrils. Are there any additional NMR resonances in the 2D and 3D spectra?”

Response:

The linewidth for ¹⁵N is about 1.5ppm; the linewidth for ¹³C is about 0.7-0.9ppm. We added the information in the paper. And we nearly assigned all the peaks in the spectra. In figure 3 (upper panel) ¹³C-¹³C 2D NMR spectrum, there are a peak assigned to Threonine (62, 70 ppm). Threonine residues are not found in the fibril core domain, but at the immediate flanking region around the core sequence. Additional broad peaks were also found. They are at around (32, 56 ppm), (20, 50 ppm).

Comment 5:

“Authors should use a mixture of the samples (¹³C/¹⁵N-labeled and unlabeled) to make sure that the cross-peaks they observed (described in 164 – 166 in the manuscript) are from intramolecular interactions.”

Response:

Thanks for the suggestion. We have prepared the RIPK3 fibrils with equimolar-mixed ^{13}C -labeled and ^{15}N -labeled protein. For this sample, the ^{13}C - ^{13}C correlation spectra gave similar cross-peaks as the fully ^{13}C -labeled one, indicating the cross-peaks described in 164-166 in the manuscript are from intramolecular interactions. We added the information to the paper at the part after line 166. A spectrum comparison was also put in the supplement figure S4 and was shown below.

Figure S4 2D ^{13}C - ^{13}C DARR spectrum of equimolar-mixed ^{13}C -labeled and ^{15}N -labeled mouse RIPK3 fibrils with a mixing time of 200 ms (blue) and the spectrum taken using the same parameters for uniformly labeled mouse RIPK3 fibril (red).

We also added a table showing all the experiments and samples used in this research.

Table S1. Experimental parameters in SSNMR for mRIPK3 fibril structure determination.

sample	MAS (KHz)	Experiment	Recycle delay	tppm (kHz)	Mixing time	Cross- polarization	Number of scans	Total Time (h)	Temperature (K)
--------	--------------	------------	------------------	---------------	----------------	------------------------	--------------------	-------------------	--------------------

			(s)		(ms)	time (ms)			
	15	DARR	2	83.3	50	1.5	36	10.8	303
	15	DARR	2	83.3	200	1.5	36	11.5	303
	15	DARR	2	83.3	500	1.5	72	29.0	303
	15	2d-NCaCX	2	83.3	50	1.5,	256	7.2	303
						5.5			
	15	2d-NCaCX	2	83.3	50	1,	256	7.2	303
mRIPK3						4.5			
(uniformly ¹³ C,	15	3d-NCaCX	2	83.3	50	1.5,	64	96	303
¹⁵ N labeled)						5.5			
	15	3d-NCaCX	2	83.3	50	1.5,	64	59.5	303
						5.5			
	15	TEDOR	2	83.3	6.4	1.5	256	18.5	303
	15	TEDOR	2	83.3	8.5	1.5	256	18.5	303
mRIPK3	15	DARR	2	100	50	1.5	48	14.3	303
(2- ¹³ C glycecol)	15	DARR	2	100	200	1.5	80	25.6	303
and	15	DARR	2	100	500	1.5	96	34.8	303
mRIPK3	15	2d-NCaCX	2	100	50	1.5,	256	19.0	303
(1,3- ¹³ C						5.5			
glycecol)	15	2d-NCaCX	2	100	50	1.5,	256	19.0	303
						4.5			
	15	TEDOR	2	100	3.2,	1.5	256	18.5	303
					6.4, 8.5				
mRIPK3	15	DARR	2	100	50	1.5	64	19.0	303
(1:1 mixed	15	DARR	2	100	200	1.5	64	20.5	303
¹³ C/ ¹⁵ N	15	DARR	2	100	500	1.5	104	37.8	303
labeled)	15	TEDOR	2	83.3	6.4,8.5	1.5	256	18.5	252
	15	NhhC	2	83.3			512	23.3	252
	15	HXINEPT	2	83.3			128	20.4	303

Comment 6:

“How many ThT fluorescence experiments were carried out in Figure S7 (errors should be indicated). Experimental conditions for fibril formation (protein concentration, incubation times etc.) should be described in Figure S7.”

Response:

We repeated the experiments three times and the error bars were added. We also added the experimental conditions in the figure S7 caption as requested. The experimental conditions were described in line 544-552 in the original manuscript. ThT was at a concentration of 50 μM and protein fibrilization was carried out by diluting 2mM protein stock solution in 6M GuDHCl to a final concentration of 20 μM in a 200 μL volume using 10mM PB buffer (pH7.4). The data were collected by measuring fluorescence intensity continuously for about 4 hours at room temperature and were plotted using Origin2018.

Figure S1 Fluorescence intensity increases during the fibril growth. The experiment was repeated three times. In this experiment, 2mM protein stock solution in 6M GuDHCl was diluted to a final concentration of 20 μM in a 200 μL volume using 10mM PB buffer (pH7.4). The fluorescence was measured immediately after the mixing

Reviewer #3 (Remarks to the Author):

The manuscript of Wu, Hu et al. seeks to bridge a major gap in understanding relating to how RIP3 can assemble into high molecular weight platforms for necroptosis signaling via the short motif termed the RHIM that is known to form amyloid fibrils. One of the mysteries arising from the Mompean et al. Cell 2018 paper was if RIP1 and RIP3 prefer an alternating arrangement, how does RIP3 self-assemble and auto-activate? The structure reported in the present submission provides some answers for this conundrum. The ssNMR structure and accompanying mutagenesis and MD indicate that, in mouse RIP3, the RHIM motif adopts an alternative conformation and relies on flanking sequences to adopt an "N" conformation. This connects the dots between prior reports (some of which should be referenced by the authors), but would benefit from validation of two key facets: (1) is the observation more broadly applicable to other species' RIP3 e.g. in human cells and (2) stronger support for mutations flanking the core RHIM tetrad impacting necroptosis in cellular studies.

Major points

1. The points of difference with earlier reported RHIM structures are not well communicated. The comparative structure figures are shown in supplement, but should be in the main text. Likewise the authors should present a sequence alignment in the main text and comment on the sequence conservation of the key residues. If they are non-homologous, further MD is warranted to examine whether a comparable structure might be observed in human RIP3. In the abstract, it would be beneficial for the authors to describe how the presented structure differs from the Mompean structure rather than simply saying they differ.
2. Relatedly, whether the observed structure would be present in other species, such as humans, is not clear from the manuscript. It would be of less interest if the structure were confined to mouse proteins and it is the species difference that is the basis for the differences with the Mompean structure. This is a crucial point that the authors need to address.
3. In relation to the differences in species: differences between the necroptosis pathway have been observed in other studies, especially between mouse and human signaling e.g. Petrie Nature Commun 2018, Tanzer CDD 2016, Davies CDD 2018, Davies Nature Commun 2020. It is important that the authors consider these differences in the context of their findings. These studies highlight the importance of establishing whether the RIP3 conformation reported is specific to mice, or whether it is illustrative of a widely conserved mechanism. Much of the introductory literature pertains to human necroptosis signaling (refs. 5, 7, 8) with incomplete coverage of the relevant mouse literature (e.g. ref. 6 Rodriguez CDD is pre-dated by Murphy Immunity 2013 and Tanzer Biochem J 2015), which seems like an oversight considering this manuscript is solely focused on the mouse system.
4. The coverage of relevant literature in the introduction is very thin and some of the ideas presented are either wrong or outmoded. This is surprising considering the seminal work

of one of the corresponding authors in the necroptosis field. For example, the authors propose the RIP1:RIP3 platform is called a signalosome. This is not the convention; it is typically called a necrosome. The authors describe the assembly of MLKL into higher order complexes via coiled coils. This is an outmoded idea. There are structural studies of MLKL that provide a clearer picture of how these assemble. The coverage of RHIM knowledge is far from complete also – e.g. Pham EMBO Rep 2018 is overlooked. Line 39- RHIM longhand is incorrect. I feel it is important that the authors familiarize themselves with the current state of the field and present a more contemporary view of the pathway. I realise it is a fast moving area but there are numerous authoritative reviews that should have led the authors to a more contemporary take on the pathway.

5. Cellular studies. The authors use an overexpression system to examine the effects of RIP3 mutants in NIH-3T3 cells in addition to some coexpression studies of RIP1 and RIP3 in 293T.

I have three major concerns with these analyses:

a. NIH3T3 cells are necroptotic when treated with TSZ (e.g. Müller et al., CMLS 2017 and the cited Sun et al. Cell 2012 on which the co-corresponding Huayi Wang is an author), which is contrary to what is stated in the manuscript. The control experiment in which untransduced NIH3T3 cells are treated with TSZ would have revealed this. I am unclear why He Cell 2009 was cited in reference to NIH3T3 cells. I would caution against this approach because examining mutant RIP3 when endogenous RIP3 is present could lead to artefacts. Examining reconstitution of necroptosis by wild-type RIP3 and the RHIM mutants in a relevant cell type like RIP3 KO mouse fibroblasts would be more compelling.

b. 293T was used to test RIP1:RIP3 interaction but this is not a necroptotic cell line. It would be more robust to express RIP3 mutants in RIP3 KO mouse fibroblasts or L929 cells, which would be consistent with some of the more reliable studies in the literature. Such an approach would be the ideal platform to examine interaction of the tagged RIP3 wild-type or extended RHIM mutant with the endogenous RIP1.

c. The authors do not examine the effect of RIP3 mutation on interaction with RIP1 in cells. While not a necroptotic line, 293T would lend themselves to such a study because it would obviate the role of RIP1 in nucleating higher order assemblies and thus RIP1 would not

contribute to any RIP3 higher order species. This would require coexpression of wild-type RIP3 and the mutants different tags for IP studies, which would be very feasible.

6. Relatedly, on line 245 the authors state that RIPK1 and MLKL are ubiquitously expressed in commonly used cell lines. This is a misrepresentation of the reality. Those used for necroptosis are used because of their ubiquitous RIP1, RIP3 and MLKL expression; not all cell lines are appropriate for these studies. 293T for example are not appropriate and arguably the most commonly used line in all of the literature. I suggest the authors rephrase this section to more accurately capture which cell types are suitable for necroptosis studies.

7. Reproducibility. The authors do not state how many repeats each experiment represents in their legends. The authors present viability data as mean and "SD of duplicate wells". Duplicates are not sufficient to enable statistical analyses and I would expect errors to represent the variation between multiple experiments.

8. In the structure figures, the authors should label the features described in the main text, such as beta-arch, in addition to the strands of interest.

9. Line 297. The authors do not validate that RIP1:RIP3 interaction are required for necroptosis, unlike what is claimed.

10. The role of the kinase domains in RIP1 and RIP3 in dictating the higher order assemblies and how these might contribute to regulation of necrosome/RHIM mediated oligomers is not considered but should be an integral part of any model

11. Line 382 and related figures. Because the colP interaction data are qualitative, the authors have not ruled out a kinetic defect in RIP1 interaction with the flanking RHIM RIP3 mutants, so this requires some caution in interpretation. Alternatively, the authors should exclude a defect in interaction/kinetic assembly by performing time course experiments.

Minor points

The manuscript would benefit from critical proofreading and/or English editing. There are many errors that distracted me during reading the work.

I find it strange that the manuscript is submitted formatted for another publisher. Regardless, the STAR Methods table is incomplete because the catalog numbers of antibodies are an important consideration when many in the field are not fit for purpose.

Figure 1E label. White text would allow the E label to be more readily seen.

Line 135. What was protein labelled with?

Line 183. Name valines.

Figure 7 legend. Software for MD needs to be described.

Answers to Reviewer #3:

Comment 1:

“The points of difference with earlier reported RHIM structures are not well communicated. The comparative structure figures are shown in supplement, but should be in the main text. Likewise the authors should present a sequence alignment in the main text and comment on the sequence conservation of the key residues. If they are non-homologous, further MD is warranted to examine whether a comparable structure might be observed in human RIP3. In the abstract, it would be beneficial for the authors to describe how the presented structure differs from the Mompean structure rather than simply saying they differ.”

Response:

We moved the supplement figure S1 to figure 1 which has the sequence comparison between major proteins containing RHIM domains and the current available structures. Additional description is provided in the abstract and the introduction.

Comment 2:

“Relatedly, whether the observed structure would be present in other species, such as humans, is not clear from the manuscript. It would be of less interest if the structure were confined to mouse proteins and it is the species difference that is the basis for the differences with the Momean structure. This is a crucial point that the authors need to address.”

Response:

This manuscript is focused on mouse RIPK3. We have also studied the structure of human RIPK3 amyloid fibrils. It turned out that the fibril structures for both human and mouse RIPK3 are similar as “N” shape. The structural studies for human RIPK3 amyloid fibrils will be reported soon.

Comment 3:

“In relation to the differences in species: differences between the necroptosis pathway have been observed in other studies, especially between mouse and human signaling e.g Petrie Nature Commun 2018, Tanzer CDD 2016, Davies CDD 2018, Davies Nature Commun 2020. It is important that the authors consider these differences in the context of their findings. These studies highlight the importance of establishing whether the RIP3 conformation reported is specific to mice, or whether it is illustrative of a widely conserved mechanism. Much of the introductory literature pertains to human necroptosis signaling (refs. 5, 7, 8) with incomplete coverage of the relevant mouse literature (e.g. ref. 6 Rodriguez CDD is pre-dated by

Murphy Immunity 2013 and Tanzer Biochem J 2015), which seems like an oversight considering this manuscript is solely focused on the mouse system.”

Response:

Thanks for the comment. As we mentioned in the last comment, the fibril structures for both human and mouse RIPK3 are similar as “N” shape. Although there are species differences between human and mouse on RIPK3-MLKL binding and MLKL activation, no difference between human and mouse was found in TNF-induced RHIM-dependent amyloid formation and RIPK3 activation. As a supporting evidence (figure shown below), we found that both wildtype mouse RIPK3 and chimaeric mouse RIPK3 with human RIPK3 RHIM (**mRIPK3_{1-327aa}hRIPK3_{323-518aa}**) could reconstituted necroptosis in L929-*Ripk3* KO cells. It means that the mouse RIPK1 RHIM could transduce upstream necroptosis signal to either human or mouse RIPK3 RHIM. There are no species differences in RHIM-dependent necroptosis signaling. We also added more references to the paper as suggested.

The L929(*Ripk3*-KO) cells stably expressing mouse RIPK3 protein by lentivirus infection were stimulated with T/Z for 10 hours. The number of surviving cells were analyzed by measuring ATP levels using Cell Titer-Glo kit. The data are represented as the mean \pm standard deviation (SD) from three independent experiments.

Comment 4:

“The coverage of relevant literature in the introduction is very thin and some of the ideas presented are either wrong or outmoded. This is surprising considering the

seminal work of one of the corresponding authors in the necroptosis field. For example, the authors propose the RIP1:RIP3 platform is called a signalosome. This is not the convention; it is typically called a necrosome. The authors describe the assembly of MLKL into higher order complexes via coiled coils. This is an outmoded idea. There are structural studies of MLKL that provide a clearer picture of how these assemble. The coverage of RHIM knowledge is far from complete also – e.g. Pham EMBO Rep 2018 is overlooked. Line 39- RHIM longhand is incorrect. I feel it is important that the authors familiarize themselves with the current state of the field and present a more contemporary view of the pathway. I realise it is a fast moving area but there are numerous authoritative reviews that should have led the authors to a more contemporary take on the pathway.”

Response:

Thanks for these suggestions.

1.” For example, the authors propose the RIP1:RIP3 platform is called a signalosome. This is not the convention; it is typically called a necrosome.”

The “***signalosome***” has been used in Mompean’s paper, which emphasized the function of RIPK1 and RIPK3 amyloid complexes is to transduce cell death signal including necroptosis and RIP1-dependent apoptosis. Considering that our manuscript mainly focused on necroptosis. We changed to “necrosome” in the revised manuscript as reviewer suggested.

2. “The authors describe the assembly of MLKL into higher order complexes via coiled coils.”

Sorry for the misleading description. The original description in old manuscript is “**Upon phosphorylation, MLKL would adopt significant structural transformation and further oligomerize, with its positively charged N-terminal coiled-coil region targeting the negatively charged phospholipids in the cellular membrane to form a pore, disrupting the plasma membrane**”. It means that after MLKL oligomerization, the

MLKL N-terminal region (containing the conserved coiled-coil domain) would bind to negatively charged phospholipids and form membrane-damaging pores. As this manuscript focused on RIP3 RHIM amyloid structure and function. We simplified the misleading description of MLKL function to make it clearer in revised manuscript as **“Upon phosphorylation, MLKL is activated and oligomerizes to disrupt the plasma membrane and cause cell necrosis”**. And we add a newest review paper from Murphy for MLKL function in reference here as reviewer suggested.

3. ***“The coverage of RHIM knowledge is far from complete also – e.g. Pham EMBO Rep 2018 is overlooked.”***

The papers the reviewer mentioned mainly describe the studies of an inhibitory viral RHIM-containing protein M45. We also published a research paper for M45 RHIM (Hu et al CDD 2020). We added both of the paper in reference in introduction part of the revised manuscript. Thanks again for the reminding.

4. ***“Line 39- RHIM longhand is incorrect.”***

It was corrected.

Comment 5:

“Cellular studies. The authors use an overexpression system to examine the effects of RIP3 mutants in NIH-3T3 cells in addition to some coexpression studies of RIP1 and RIP3 in 293T. I have three major concerns with these analyses:

a. NIH3T3 cells are necroptotic when treated with TSZ (e.g. Müller et al., CMLS 2017 and the cited Sun et al. Cell 2012 on which the co-corresponding Huayi Wang is an author), which is contrary to what is stated in the manuscript. The control experiment in which untransduced NIH3T3 cells are treated with TSZ would have revealed this. I am unclear why He Cell 2009 was cited in reference to NIH3T3 cells. I would caution against this approach because examining mutant RIP3 when endogenous RIP3 is present could lead to artefacts. Examining reconstitution of necroptosis by wild-type RIP3 and the RHIM mutants in a relevant cell type like RIP3

KO mouse fibroblasts would be more compelling.

b. 293T was used to test RIP1:RIP3 interaction but this is not a necroptotic cell line.

It would be more robust to express RIP3 mutants in RIP3 KO mouse fibroblasts or L929 cells, which would be consistent with some of the more reliable studies in the literature. Such an approach would be the ideal platform to examine interaction of the tagged RIP3 wild-type or extended RHIM mutant with the endogenous RIP1.

c. The authors do not examine the effect of RIP3 mutation on interaction with RIP3 in cells. While not a necroptotic line, 293T would lend themselves to such a study because it would obviate the role of RIP1 in nucleating higher order assemblies and thus RIP1 would not contribute to any RIP3 higher order species. This would require coexpression of wild-type RIP3 and the mutants different tags for IP studies, which would be very feasible.”

Response:

1. “a. NIH3T3 cells are necroptotic when treated with TSZ (e.g. Müller et al., CMLS 2017 and the cited Sun et al. Cell 2012 on which the co-corresponding Huayi Wang is an author), which is contrary to what is stated in the manuscript. The control experiment in which untransduced NIH3T3 cells are treated with TSZ would have revealed this. I am unclear why He Cell 2009 was cited in reference to NIH3T3 cells..”

NIH 3T3 cells typically do not have the western-detectable RIPK3 expression, which was extensively studied in **Zhang et al., Science 2009**, one of the three milestone papers identified the RIPK3 as the key regulator of necroptosis. The related descriptions and figure in Zhang’s paper are listed here. **“Although NIH 3T3 cells typically undergo apoptosis in response to tumor necrosis factor (TNF) stimulation, caspase-independent cell death has been reported in one NIH 3T3 line (termed N cells here). We compared the caspase-dependence of TNF-induced cell death between NIH 3T3 cells obtained from America Type Culture Collection (termed A cells here) and N cells.”** **“Differing expression of RIP3 in A and N cells was confirmed by means of Western blotting (Fig. 1C)”**, **“RIP3 expression in A cells switched TNF-induced apoptosis to caspase-**

independent cell death (Fig. 1E)". (We attached the original figure Fig 1C and 1E below)
 The NIH 3T3 cells used in this study or **He et al., Cell 2009** we cited in this manuscript or **Sun et al., Cell 2012** the reviewer mentioned, belong to the same cell strain originally obtained from ATCC and do not have western-detectable RIPK3 expression. Moreover, in **Sun et al, Cell 2012**, NIH3T3 cells were only used in Figure S1E and S7A for exogenous expression of RIPK3. When NIH3T3 cells were transfected with control plasmid (vector), they were not necroptotic treated with TSZ (Figure S7A, Sun et al. Cell 2012). We will add **Zhang et al., Science 2009** to the reference of the NIH 3T3 system .

Zhang et al., Science 2009: Fig. 1C and 1E

2. "would caution against this approach because examining mutant RIP3 when endogenous RIP3 is present could lead to artefacts. Examining reconstitution of necroptosis by wild-type RIP3 and the RHIM mutants in a relevant cell type like RIP3 KO mouse fibroblasts would be more compelling."

Endogenous RIPK3 is not present in the cell in our study as we described in the last comment. But we have examined the RIPK3 mutants in MEF(*Ripk3*-KO) cells as reviewer

suggested. The results were similar as in NIH-3T3 cells (Figure S6B in the revised manuscript)

B

Figure S6B The MEF(*Ripk3*-KO) cells stably expressing mouse wild and mutated RIPK3 protein by lentivirus infection were stimulated with T/S/Z for 10 hours. The number of surviving cells were analyzed by measuring ATP levels using Cell Titer-Glo kit. The data are represented as the mean \pm SD of duplicate wells. The mouse RIPK3 expression level was measured by western blot analysis. All experiments were repeated three times.

3. ***“b. 293T was used to test RIP1:RIP3 interaction but this is not a necroptotic cell line. It would be more robust to express RIP3 mutants in RIP3 KO mouse fibroblasts or L929 cells, which would be consistent with some of the more reliable studies in the literature. Such an approach would be the ideal platform to examine interaction of the tagged RIP3 wild-type or extended RHIM mutant with the endogenous RIP1.”***

The Co-IP of RIPK1-RIPK3 in 293 cells just measures the soluble-interactions of RIPK1 and RIPK3, which do not need necroptosis induction and do not prevented by RIPK1 kinase inhibitors. Co-IP in 293 cells was used to identify the RHIM domain as a novel self-interaction domain in RIPK1 and RIPK3 (Sun et al., JBC 2002) or in other upstream proteins such as TRIF (Meylan et al., Nat Immunology 2004). Meanwhile in necrotic cells, including HT-29 or MEF, soluble RIPK1-RIPK3 interactions are only detected upon necroptosis-induction and could be repressed by RIPK1 inhibitor such as Nec-1. Since the fibrillar amyloids are protein aggregates which could not be soluble by regular lysis buffer triton even added with 2 - 4 M urea, the IP of RIPK1 and RIPK3 in necrotic cells upon necroptosis-induction only detects the soluble transition state of RIPK1-RIPK3 hetero-

amyloids. It is different from co-IP in HEK293T cells. We have tried MEF cells as suggested and found all the RIPK3 could not co-IP with RIPK1.

The MEF(*Ripk3*-KO) cells stably expressing mouse wild and mutated RIPK3 protein by lentivirus infection. Whole-cell lysates were subjected to immunoprecipitation with anti-Flag M2 beads. The total cell lysates and immunoprecipitates were immunoblotted with the indicated antibodies.

4. “c. The authors do not examine the effect of RIP3 mutation on interaction with RIP3 in cells. While not a necroptotic line, 293T would lend themselves to such a study because it would obviate the role of RIP1 in nucleating higher order assemblies and thus RIP1 would not contribute to any RIP3 higher order species. This would require coexpression of wild-type RIP3 and the mutants different tags for IP studies, which would be very feasible.”

We did the Co-IP assay of wild-type and mutant form of RIPK3 as reviewer suggested. It showed that L456D binds to WT RIPK3 well, and the binding of F442D to WT is attenuated while other mutants hardly bind to WT RIPK3. It suggested changes in different parts of RIPK3 RHIM affect self-assembly differently.

The HEK293T cells were co-transfected with DNA plasmids containing mouse Myc-tagged RIPK3 and Flag-tagged mouse RIPK3 (or its mutants). Cell lysates were collected 36h post-transfection, and immunoprecipitated with anti-Flag magnetic beads (Bimake) at 4 °C. The total cell lysates and immunoprecipitates were analyzed by western-blot analysis with the indicated antibodies.

Comment 6:

“Relatedly, on line 245 the authors state that RIPK1 and MLKL are ubiquitously expressed in commonly used cell lines. This is a misrepresentation of the reality. Those used for necroptosis are used because of their ubiquitous RIP1, RIP3 and MLKL expression; not all cell lines are appropriate for these studies. 293T for example are not appropriate and arguably the most commonly used line in all of the literature. I suggest the authors rephrase this section to more accurately capture which cell types are suitable for necroptosis studies.”

Response:

Thanks for the reminding, this description came from the studies in **He et al., Cell 2009**. In this paper, 14 commonly used cell lines were tested TSZ-induced necroptosis and RIPK3 expression. And they found **“The Expression of RIP3 Correlates with Cellular**

Necrotic Response". We will change this description to **"TNF-induced necroptosis is mediated by RIPK1, RIPK3 and MLKL. Interestingly, ectopic expression of a functional RIPK3 can convert necroptosis-resistant cells such as mouse NIH-3T3 or human HeLa cells to sensitive ones"** in the revised manuscript to avoid misleading.

Comment 7:

"Reproducibility. The authors do not state how many repeats each experiment represents in their legends. The authors present viability data as mean and "SD of duplicate wells". Duplicates are not sufficient to enable statistical analyses and I would expect errors to represent the variation between multiple experiments."

Response:

Three or more repeated experiments with each duplicated were done for the cell studies and were originally stated in the method section. We added the information to the legends.

Comment 8:

"In the structure figures, the authors should label the features described in the main text, such as beta-arch, in addition to the strands of interest."

Response:

We modified the figure showing the SSNMR structure of RIPK3 fibrils, adding the necessary information (figure 6).

Comment 9:

"Line 297. The authors do not validate that RIP1:RIP3 interaction are required for necroptosis, unlike what is claimed."

Response:

Thanks for the reminding, we revised the improper description as “ **TNF-induced necroptosis pathway requires RIPK1-RIPK3 intermolecular interaction. Meanwhile the functional studies above indicate that the downstream necroptotic process could not be activated without the correct RIPK3 self-assembly formation.**” in the revised manuscript.

Comment 10:

“The role of the kinase domains in RIP1 and RIP3 in dictating the higher order assemblies and how these might contribute to regulation of necrosome/RHIM mediated oligomers is not considered but should be an integral part of any model.”

Response:

There is a long linker region between the RHIM and kinase domain of RIPK1 and RIPK3. The kinase domains are probably outside the RHIM assemblies. Besides that, in **Li et al Cell 2012**, they found the overexpression of truncated RIPK1/3 containing RHIM (RIPK1_496-583 and RIPK3_388-518), but not full-length RIPK1/3 could form amyloid puncta without necroptosis induction (Figure S7B in **Li et al Cell 2012**). It suggested RHIM self-assembly was repressed by either the linker region or the kinase domain of RIPK1/3 itself directly or indirectly through unknown proteins binding to these regions. So that, it is important to study the regulation process of full-length RIPK3 amyloid formation in the future, but it is beyond our purpose of this study.

Comment 11:

“Line 382 and related figures. Because the colP interaction data are qualitative, the authors have not ruled out a kinetic defect in RIP1 interaction with the flanking RHIM RIP3 mutants, so this requires some caution in interpretation. Alternatively, the authors should exclude a defect in interaction/kinetic assembly by performing time course experiments.”

Response:

Thanks for the comment, When the cellular RHIM containing proteins form fibrillar amyloids, they will translocate to insoluble fraction and became urea resistant aggregates. So that the co-IP just measure the soluble RIPK1 binding to RIPK3, a transition stage of RIPK1/RIPK3 complex. It is hard to measure the kinetics of amyloid formation in cells. It is common technical challenge for not only RHIM but also other amyloid proteins. Yes, we have not ruled out a kinetic defect in soluble RIPK1-RIPK3 interaction and soluble RIPK3 self-interactions. The kinetic defect of soluble interactions may also affect the amyloid formation. But it is still supporting our model that mutations change the proper amyloid formation of RIPK3 will affect RIPK3 activation process by either structurally or kinetically.

Minor points

The manuscript would benefit from critical proofreading and/or English editing.

There are many errors that distracted me during reading the work.

I find it strange that the manuscript is submitted formatted for another publisher.

Regardless, the STAR Methods table is incomplete because the catalog numbers of antibodies are an important consideration when many in the field are not fit for purpose.

We have changed the format, added the catalog numbers in the figure legend and method. We also edited the language.

Figure 1E label. White text would allow the E label to be more readily seen.

It has been changed.

Line 135. What was protein labelled with?

Solid-state NMR was carried out with samples with different isotopically labelled proteins.

A new table S1 has summarized all the samples and experiments.

Line 183. Name valines.

The information has been added (V441, V448, and V457).

Figure 7 legend. Software for MD needs to be described.

The information has been added in the figure 7 (new figure 8 legend). The software is Xplor-NIH.

REVIEWERS' COMMENTS

Reviewer #1 (Remarks to the Author):

The questions and comments in my review have been addressed.

Reviewer #2 (Remarks to the Author):

The authors properly addressed the reviewer's concern in the revised manuscript.

Reviewer #3 (Remarks to the Author):

The authors have satisfactorily addressed my queries. I am surprised, however, that the human RIPK3 study is being published separately.